# In toto analysis of embryonic organisation reduces tissue diversity to two archetypes requiring specific cadherins

Max Brambach [1,2] ✉, Jana Wittmann [1], Marvin Albert [1,3], Jérôme Julmi[1], Robert Bill [1] & Darren Gilmour [1] ✉

Organisms are far greater than the sum of their differentiated cells, as the function of most cell types emerges from their organisation into three-dimensional tissues. Yet, the mechanisms underlying architectural diversity remain poorly understood, partly due to a lack of methods for directly comparing different tissue organisations. Here we establish nuQLOUD, an efficient imaging and computational framework that reduces complex tissues to clouds of nuclear positions, enabling the extraction of cell-type agnostic architectural features. Applying nuQLOUD to whole zebrafish embryos reveals that global tissue diversity can be efficiently reduced to two archetypes, termed 'amorphous' and 'crystalline'. We investigate the role of cadherin cell adhesion molecules in controlling organisational diversity and demonstrate that their expression segregates along tissue-archetypal lines. Targeted perturbations identify N-cadherin as a general driver of the amorphous archetype. This organisation-centric approach provides a way to conceptualise tissue diversification and investigate the underlying mechanisms within a standardised, quantitative framework.

Cells are the basic units of life. Advances in transcriptomics now enable cellular diversity within organisms to be characterised at an unprecedented resolution[1–3]. The power of this approach is that it exploits the standard language of mRNA sequences to define and compare diverse cells in a systematic and quantitative manner. When combined with perturbation experiments, such expression profiling methods can identify drivers of cell type diversification[4,5]. However, characterising cell compositions via current -omics methods is only part of the equation, as organisms are much more than the sum of their cell types. Indeed, the biological functions of most cell types only emerge when they are organised into multicellular tissues with specific architectures[6]. The barrier function of the skin depends on keratinocytes being organised in sheets, the contractile function of muscles depends on their being bundled into longitudinal fibres, the computational function of the brain depends on neurons being connected in networks, and so on. Tissues can therefore be considered the

functional units of life, in metazoans at least, and these functions require proper organisation. Common frameworks for the quantification of tissue organisation could provide a basis for examining how evolving structural features of tissues enable their specific functions.

An important step towards characterising architectural diversity is the development of advanced imaging methods such as light-sheet microscopy, which allow standardised imaging of whole developing organisms at single-cell resolution[7,8]. Such *in toto* imaging has been combined with large-scale tracking of cell nuclei to provide insights into dynamic germ layer interactions during zebrafish gastrulation[9], the origins of cell fates from gastrulation to organogenesis stages of mammalian development[10] and several other important aspects of development[11–14]. However, while *in toto* microscopy enabled unprecedented imaging of entire animals, this breakthrough has not yet delivered a greater understanding of the architectural diversity that exists within organisms. A key limitation has been a lack of standardised

[1]Department of Molecular Life Sciences, University of Zurich, Zürich, Switzerland. [2]Present address: Harvard Medical School, Boston, MA, USA. [3]Present address: Institut Pasteur, Paris, France. ✉e-mail: max_brambach@hms.harvard.edu; darren.gilmour@uzh.ch

frameworks for quantifying and comparing multicellular organisation. Bioimage analysis has generally focused on single-cell segmentation methods that prove to be very powerful for the description of individual cell morphologies[15,16]. However, it can be challenging to extend these to complex tissues and whole organisms due to the high variability in cell morphologies and distributions[17]. This heterogeneity also underlies another challenge: descriptors that can define the shape of cells in one tissue may not be informative for another. For example, epithelial cells can often be quantified as efficiently packed polygons[18], whereas ongoing efforts to define the architecture of the nervous system focus on the morphology and connectivity of axonal and dendritic protrusions[19]. Cell morphology segmentation methods encourage a focus on cell-scale features that may not be relevant to architecture at the tissue scale. Thus, the development of efficient frameworks that allow diverse tissue organisations to be directly compared in a 'cell-type agnostic' manner represents an important step towards understanding the architectural diversity that exists within and across organisms. The lack of methods for comparing multicellular architecture has specifically hindered investigations into the genetic regulation of tissue diversification. At the cellular scale, morphogenesis is driven primarily by the actomyosin cytoskeleton and regulated by RhoA GTPases universally present in all cells. At the tissue scale, cytoskeletal activities are interconnected via cell-cell adhesion that coordinate cell shaping and generate specific multicellular structures required for diverse organ functions. This indicates that patterned expression of cell adhesion molecules, most notably cadherins, could enable genetically-regulated changes in tissue organisation. Cadherins have long been studied for their role in tissue integrity and changes in cadherin expression are hallmarks of degenerative diseases and cancer[20,21]. Differential expression of cadherins can promote tissue segregation during early development[22] and cell sorting in 'synthetic' embryos[23]. Beyond segregation, patterned cadherin expression plays a key role in coordinated cell movements[24,25] and lately a 'cadherin code' has been shown to increase the robustness of morphogen patterning in the central nervous system (CNS)[26]. Notably, the large expansion of the cadherin superfamily that coincided with the emergence of vertebrates—with more than 100 cadherin and cadherin-related proteins being present in humans[27]—has been proposed to reflect the greater morphological diversification observed in these species[28]. However, it remains unclear if differences in cadherin expression can explain the different arrangements of cells that shape tissues.

In this study, we present a general approach to quantify organisational diversity by simplifying tissue complexity to the three-dimensional arrangement of constituent cells, 'focusing on the forest rather than the trees'. By performing nuclear segmentation across entire embryos, we defined cellular positions within a common point-cloud-based framework, a standard data structure in many spatial analysis tasks, enabling comparative organisational analysis of developing tissues in a cell-type-agnostic manner. This coarse-grained approach allowed us to track tissue organisation over time and correlate organisational features with gene expression. Unbiased clustering of similarly organised cells revealed that tissue heterogeneity can be reduced to two diametrically opposed organisational 'archetypes,' which we termed 'amorphous' and 'crystalline'. Defining the *in toto* expression of 12 major cadherin cell adhesion receptors showed that cadherin expression domains become bi-partitioned along tissue archetypal lines. Further investigation using spatiotemporal correlation analysis and targeted perturbations identified N-cadherin as a general regulator of the amorphous archetype.

## Results

### nuQLOUD quantifies global tissue organisation via 3D nuclear positions

To enable the investigation of the organisational diversity of entire developing organisms, we initially focused on zebrafish embryos at 48 h post-fertilisation (hpf), a stage where the progenitors of several major organs are already formed[29]. Multi-view light sheet microscopy was used to generate isotropic resolution imaging data of whole embryos where nuclei were stained with DAPI and individual cells were localised using TGMM nuclear segmentation (Fig. 1a, Supplementary Fig. 1), robustly and accurately identifying the centre of mass of individual nuclei[10] (Supplementary Fig. 2). While nuclear segmentation is routinely used to count and track cells[9,10], here we explored its potential for quantifying tissue organisation *in toto*. We reasoned that quantifying the 3D arrangement of cells via the position of their nuclei may provide an effective and standardised method to compare conserved features of tissue architecture. Overall, we quantified the distribution of more than 4,000,000 cells in their native environment, across 34 samples, an average of $117,000 \pm 9000$ cells per embryo. We next took advantage of the explicit point-cloud structure of the data to characterise the local organisation of nuclei using a three-dimensional Voronoi diagram, a spatial partitioning algorithm that assigns each point a volume of closest 'cells' from point distributions. Such Voronoi cells are constructed by dividing 3D space into regions consisting of all points closer to a given point than to any other point. The shape and size of such Voronoi cells consequently provide an object-based representation of local, multicellular organisation[30]. A known limitation of conventional Voronoi diagrams is that cells at the embryo's edge can have infinite size due to a lack of neighbouring points to limit their extent, making the method unsuitable for boundary tissues like skin. To overcome this, we restricted the size of boundary Voronoi cells adaptively to their neighbourhood by placing artificial points around the perimeter of the embryo such that each local neighbourhood is smoothly extended by one cell layer (see Supplementary Fig. 3, Supplementary Note 1 for details). We used this adaptively restricted Voronoi diagram to characterise the organisational context of every nucleus in entire zebrafish embryos using fourteen engineered features, including kernel density estimations over different length scales, number of neighbours and the polarity of the Voronoi cell (Fig. 1b, Supplementary Fig. 4a–d, Supplementary Note 2). Clustering of these features by similarity across all analysed cells identified three organisational feature classes based entirely on local nuclear position that we termed anisotropy, density, and irregularity (Fig. 1c, Supplementary Fig. 4e, f). Individual features contributed equally to the overall variance of the data and the spatial distribution of features highlighted different anatomical regions in a combinatorial way (Fig. 1d, Supplementary Fig. 4g–i). Moreover, individual feature distributions were conserved between samples (Supplementary Fig. 5). This demonstrates that our whole-organism description of multicellular arrangement robustly captures meaningful organisational differences within one unified framework, which we term NUclear-based Quantification of Local Organisation via cellUlar Distributions (nuQLOUD).

To map out the organisational landscape of whole embryos, we used nuQLOUD to characterise and compare the organisation of major tissues, including skin, brain muscles and connective tissue (Fig. 1e, Supplementary Fig. 6). For that, we incorporated tissue specific gene expression information from transgenic reporter lines and via hybridisation chain reaction fluorescent in situ hybridisation (HCR). We classified all cells for the presence or absence of fluorescence using an adapted version of the *in toto* imaging and segmentation approach (Supplementary Fig. 7). In this way, we were able to assign sets of organisational features to genetically defined tissues Fig. 1f, Supplementary Fig. 6b) and compare these directly (Fig. 1g, Supplementary Fig. 6c). This allowed projecting tissue landmarks onto the feature space representation and indicated organisational overlap between tissues that are far apart in both lineage and function, such as skin and muscles, which are derived from ectoderm and mesoderm, respectively; both exhibit low density and a high degree of anisotropy (Fig. 1f, g, Supplementary Fig. 6c). Interestingly, clustering of all analysed tissues based on their organisation revealed a bimodal distribution with

muscles, skin and yolk syncytial layer (YSL) in one group and eyes and CNS in the other (Fig. 1f). These results show how the nuQLOUD framework can be used to identify shared organisational patterns across vastly different organs in whole organisms.

## Reducing embryonic organisational complexity to two tissue archetypes

Next, we expanded our nuQLOUD analysis from gene expression-defined organs to all cells at multiple stages of zebrafish embryogenesis to identify and analyse stereotypical organisational patterns. Cells with shared organisational features were grouped into eleven 'organisational motifs' using a Gaussian mixture model (Fig. 2a, Supplementary Fig. 8, Supplementary Fig. 9, Supplementary Note 3). The spatial distributions of individual motifs were compact and mapped to anatomical regions when projected on individual samples, highlighting that the nuQLOUD framework was able to differentiate between tissues solely based on the local distribution of nuclei

(Supplementary Fig. 10). We tracked the organisational motifs between 12 and 72 hpf and found that some motifs were present at all time points, while others were absent early and developed over time (Fig. 2a, b). One example for a maintained motif was motif IX, which consistently mapped to the YSL from 12 hpf on. However, cells of the median fin fold were classified in this motif from 48 hpf onward, highlighting an organisational similarity between these tissues (Fig. 2b IX). Motif I evolves from being essentially absent at 12 hpf, to highlighting cells of the spinal cord and hindbrain at 24 hpf before spreading to include cells of the forebrain and eyes by 48 hpf, indicating that in different CNS regions cells adopt this organisation in a temporal sequence (Fig. 2b I). This demonstrates that organisational complexity increases during early development and highlights that nuQLOUD is able to detect organisational patterns without gene expression information in an unbiased way.

To understand higher-level patterns of tissue organisation, we investigated how organisational motifs relate to each other. We used

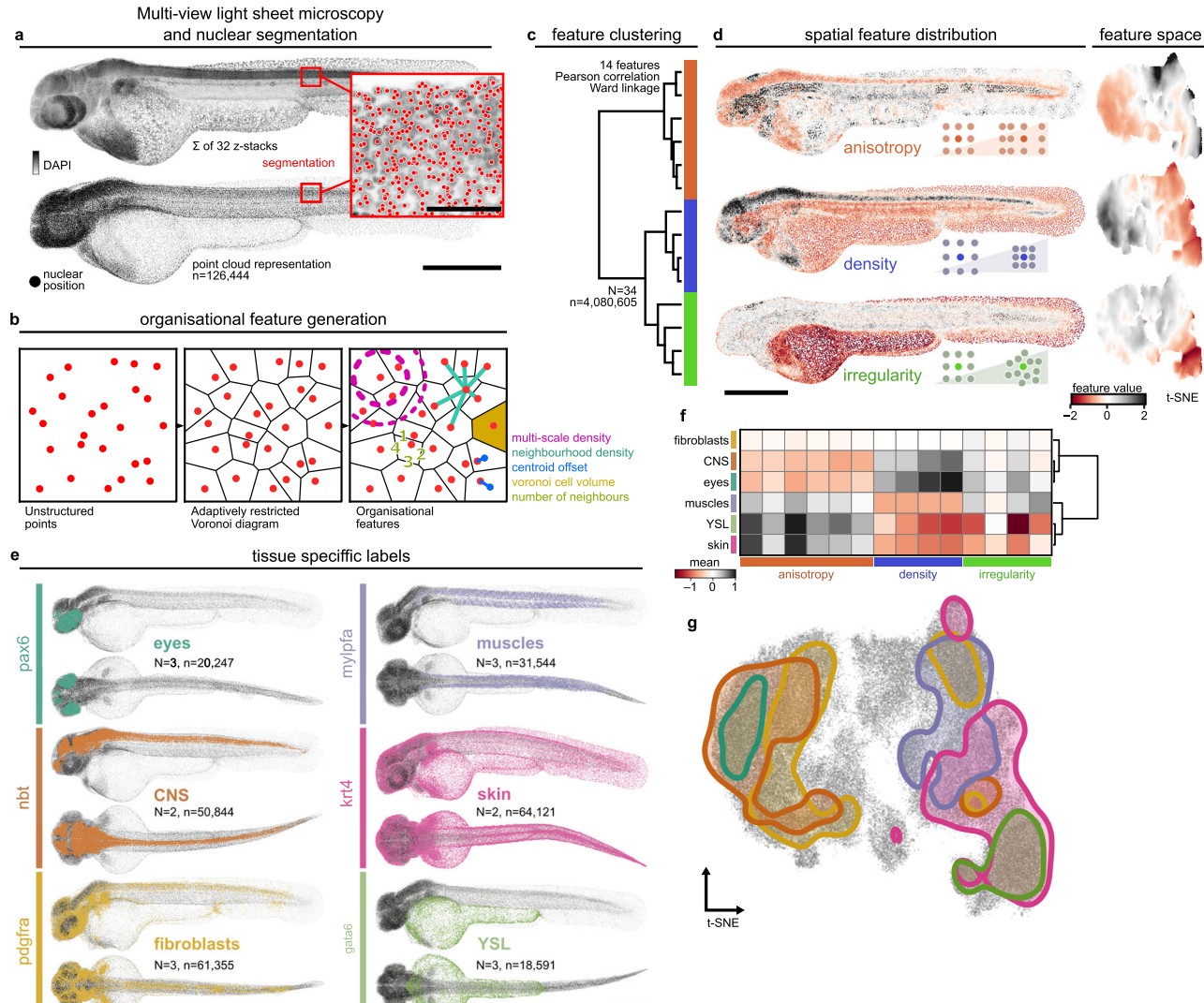

**Fig. 1 | In toto quantification of multicellular organisation. a** Multi-view light sheet microscopy and nuclear segmentation transform in toto volumetric image data into point cloud representation. Inset scale bar: 50 μm. **b** Each cell's local organisation is quantified by a 14-dimensional feature vector describing the surrounding point distribution; absolute coordinates are not included in feature set. **c** Clustering of organisational features reveals three distinct feature classes: anisotropy, density, and irregularity. **d** Feature classes show distinct anatomical distributions and occupy separate regions in organisational feature space (t-SNE

embedding). **e** Tissue-specific gene expression patterns are mapped onto the point cloud; individual nuclei are classified as expressing or non-expressing based on HCR fluorescence or transgenic reporter intensity. **f** Mean organisational feature profiles quantify organisational diversity. **g** Gene expression domains (from e) overlayed on t-SNE embedding of organisational feature space. Feature values are z-scored across all cells of multiple samples. *N* number of samples, *n* total number of cells. All scale bars except inset 500 μm. Source data are provided as a Source Data file.

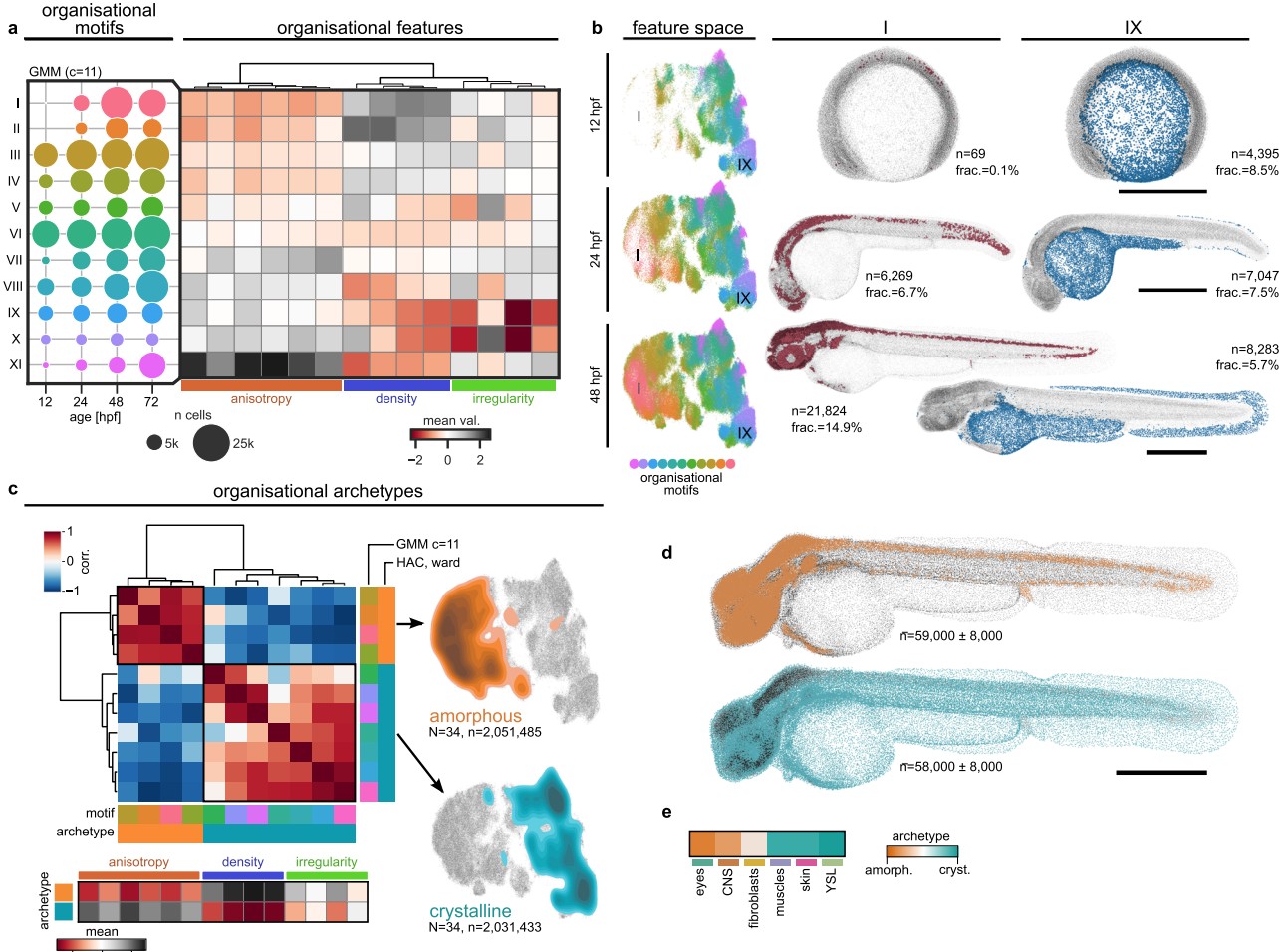

**Fig. 2 | Global organisational heterogeneity reduces to two organisational archetypes. a**, **b** Organisational diversity increases over time. A Gaussian Mixture Model (GMM) stratifies organisational feature space into eleven 'organisational motifs' using pooled data from four developmental stages. Some motifs persist throughout early development (e.g. III, VI, IX), while others emerge dynamically (e.g. I, II, VII). Feature values are z-scored; clustering is performed across all time points. **b** Example motifs: Motif I is absent early and emerges from 24 hpf on, ultimately covering the CNS; Motif IX is consistently found in the YSL and later includes the median fin fold. Feature space (t-SNE) based on all time points. *n*: number of cells per motif in the shown sample. **c**–**e** Organisational diversity

converges into two archetypes. **c** Hierarchical agglomerative clustering (HAC) of organisational motifs (48 hpf data) identifies two archetypes with opposing features. 'Amorphous' cells are dense, isotropic, and irregular; 'crystalline' cells are low-density, anisotropic, and regular. Feature values are z-scored. HAC used Euclidean distance and Ward linkage. **d** Archetypes map to cohesive spatial domains at 48 hpf. CNS, sensory organs, and fin buds are amorphous; muscles, skin, and YSL are crystalline. **e** Cells expressing tissue-specific markers align with one of the two archetypes, with the exception of fibroblasts. *n* mean number of cells per motif across *N* = 34 samples ± standard deviation. All scale bars 500 μm. Source data are provided as a Source Data file.

hierarchical agglomerative clustering with Ward linkage to group motifs on their organisational similarity. This revealed two classes, which we term organisational 'archetypes', each accounting for ~50% of cells in the embryo at 48 hpf. Based on their organisational profile, we found that the two archetypes fit with the definitions of 'crystalline' and 'amorphous' from material sciences (Fig. 2c), however we stress that this classification only refers to organisational patterns and does not imply the corresponding material properties such as rigidity and fluidity. Tissues assigned to the amorphous archetype showed a dense, isotropic and irregular arrangement of nuclei and included cells of the eyes, CNS and pectoral fin buds. By contrast, tissues assigned to the crystalline archetype exhibited nuclei arranged in a low-density, anisotropic and regular fashion and included cells of the muscles, skin and YSL (Fig. 2d, e). This bimodality becomes especially apparent when the archetypes are projected onto the embedding of the organisational feature space, with each covering an opposing hemisphere (Fig. 2c). Random forest classification-based feature importance evaluation and feature-dropout confirmed that all organisational features contributed to this classification (Supplementary Fig. 11a, b), and analysis of this

classification across a range of initial cluster numbers showed that the archetypes are robust and intrinsic to the data (Supplementary Fig. 11c–e). Moreover, we found that this bipartitioning was not a feature of a single developmental stage but could be identified throughout 12 and 72 hpf (Supplementary Fig. 11f, g). We provide three-dimensional renderings of embryos at 12, 24, and 48 hpf with the two archetypes highlighted in Supplementary Data 1–3. These results show that the bimodal organisation of bona-fide tissues identified in the previous section (Fig. 1f) extends to the whole organism and that local cellular arrangement follows one of two general organisational archetypes.

**Cadherin expression patterns align with tissue archetypes**
To identify potential genetic regulators of this organisational bipartitioning into archetypes, we investigated the cadherin family of cell adhesion receptors, important regulators of tissue integrity and differential adhesion-based cell sorting[23,31]. While much is known about how cadherins function at the molecular, cellular, and biophysical level, their impact on tissue organisational diversity of

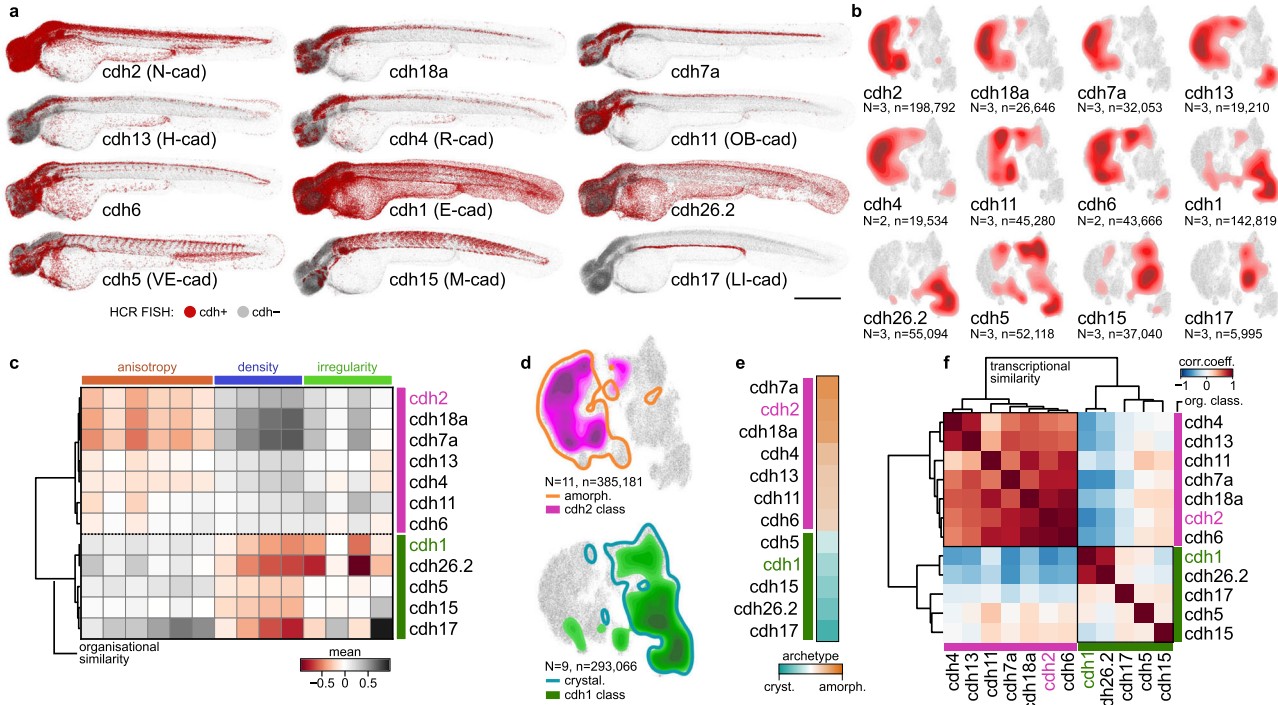

**Fig. 3 | Organisational similarity identifies two classes of cadherin-expressing cells that align with archetypes. a** Cadherin expression domains are mapped onto the point cloud representation; individual nuclei are classified as expressing or non-expressing based on HCR fluorescence intensity. Cadherins expressed in >1% of all cells were included (see Supplementary Fig. 12a). Scale bar: 500 μm. **b** Expression domains occupy distinct regions in organisational feature space, localising predominantly to either the left or right hemisphere. **c** Clustering of cadherin-expressing cells based on organisational features reveals two major classes. Mean feature values are z-scored per domain. **d** The two cadherin classes map to opposite halves of feature space and correspond predominantly to amorphous and crystalline archetypes. **e** Cadherin classes show distinct association with organisational archetypes. Colour scale: proportion of class-associating cells (cyan/orange = 100% association; white = 50% each). **f** Auto-correlation clustering of cadherin expression domains based on single cell RNA sequencing data[33] yields the same E- and N-cad-like classes as the classification based on organisational similarity. Pearson correlation coefficient. *N* number of samples, *n* total number of cells, dendrograms constructed from hierarchical agglomerative clustering using Ward linkage and Euclidean distance.

whole organisms is less clear. To examine whether cadherins play a role in this context, we probed the relationship between specific cadherin expression and architecture by projecting *in toto* expression patterns of twelve highly expressed cadherins (Supplementary Fig. 12a), defined by HCR, onto the nuQLOUD framework (Fig. 3a). This revealed that each cadherin expression domain localised to discrete, albeit overlapping regions in feature space, indicating that cells expressing each specific cadherin had a range of organisation that is definable with our approach (Fig. 3b). Clustering based on organisational similarity allowed cadherin expression domains to be grouped into two classes that mapped to the previously identified organisational archetypes; prominent members of the amorphous cadherins are N-cadherin (N-cad/cdh2) and cdh11, while members of the crystalline class include E-cadherin (E-cad/cdh1) and cdh15 (Fig. 3c–e). This clustering of cadherin family members based on the organisation of expressing tissues was not predicted by similarities in their amino acid sequence or structures[32] (Supplementary Fig. 13). Grouping cadherin expressing cells based on their transcriptional similarity as defined by single cell RNA sequencing[33] generated a subdivision of cadherin family members into the same classes (Fig. 3f, Supplementary Fig. 12b, c). Pairwise co-expression analysis ruled out single-cell co-expression of multiple cadherins as reason for the organisational and transcriptional bimodal classification (Supplementary Fig. 12d). By contrast, comparing cadherin expression profiles with those of the 'matrisome', a curated panel of some 900 extracellular matrix (ECM) proteins and regulators[34], did not identify ECM components whose expression segregates along tissue archetypal lines (Supplementary Fig. 14). The correlation between cadherin expression patterns and archetypal organisation suggested

that individual cadherins may be required for the organisation of their respective expression domains. To test that, we systematically mutated individual cadherins using CRISPR/Cas9, a strategy that proved to effectively edit the respective loci as confirmed by sequencing (Supplementary Fig. 15a, b). We specifically analysed the organisational effect on cells that normally would express the targeted cadherin, which we identified via HCR FISH (Supplementary Fig. 15c). While cadherins are known to act redundantly[35], loss of the amorphous class cadherins cdh7a, cdh13 and cdh18a, resulted in their target tissues shifting towards the crystalline archetype (Supplementary Fig. 15d–f). As loss of E-cad results in an embryonic lethal phenotype, we applied a genetic mosaic approach to address the requirement for E-cad in the crystalline organisation of the basal cell layer of the epidermis (Supplementary Fig. 16a–c). nuQLOUD analysis of Ecad-deficient basal cell clones, identified via loss of an endogenously-tagged reporter (Cdh1:Cdh1-YFP) revealed a significant shift towards amorphous organisation (Supplementary Fig. 16d–f), a shift that occurred without upregulation of N-cad (Supplementary Fig. 16g). Thus, loss of function experiments revealed that the cadherin classes identified by nuQLOUD organisational analysis play important roles in promoting their respective tissue archetypes. Moreover, these data on cadherins highlight how nuQLOUD can be used to highlight connections between transcriptional and tissue-organisational patterns; links that can then be tested with perturbations using the same quantitative framework.

**N-Cad expression dynamics mirror tissue archetype-switching**
To investigate the link between cadherin expression and organisational archetypes further, we reasoned that dynamic changes in

cadherin expression should correlate with corresponding organisational changes over time. We therefore selected E- and N-cad as representatives of the amorphous and crystalline organising cadherin classes and tracked their expression patterns using transgenic reporters (TgCRISPR(cdh1-mLanYFP)[36], TgBAC(cdh2:cdh2-GFP))[37] between 12 and 72 hpf (Fig. 4a). E-cad expression remains high in the epidermis and other epithelial cells that maintain a crystalline organisation throughout. By contrast, the pattern of N-cad expression is dynamic, being expressed in most embryonic cells at 12 hpf, before becoming focused to cells of the CNS and a subset of other tissues from 48 hpf onwards (Fig. 4a–c, Supplementary Fig. 17). N-cad positive cells were assigned to the amorphous archetype across the analysed time window; an association that was agnostic to cell type. For example, mesenchymal cells of the pectoral fins and neurons of the brain have little commonality in lineage or function but share amorphous organisation and N-cad expression (Fig. 4b). This dynamic link between E-/ N-cad expression and archetypal tissue organisation further supports the idea that specific cadherin expression regulates local tissue organisation.

To test this in more detail, we next addressed this relationship within a single tissue, focusing on the developing fast-twitch muscles between 12 and 48 hpf, a well-studied morphogenic system that exhibits dynamically changing cadherin expression, where N-cad is expressed homogeneously early before becoming restricted to a superficial layer of slow muscle cells later[38]. We sub-selected developing muscles based on their nuclear morphology and tracked their cadherin expression at key time points, which confirmed that muscle cells inside each forming somite downregulate N-cad expression in an anteroposterior 'wave-like' manner (Fig. 4d, Supplementary Movies 1–3). nuQLOUD analysis revealed that fast twitch muscle progenitors switch their organisational archetype from amorphous at 12 hpf to crystalline at 48 hpf (Fig. 4e). At 24 hpf this organisational transition shows a similar pattern to N-Cad expression within the same tissue: posterior cells expressed N-cad and remained in an amorphous organisation whereas anterior cells were N-cad non-expressing and showed crystalline organisation (Fig. 4e). In the transition zone midway along the anteroposterior axis we identified a population of N-cad non-expressing, amorphously organised cells, but very few N-cad expressing crystalline cells (Fig. 4f), suggesting that cells downregulate their N-cad expression prior to their archetypal transition. Moreover, expression analysis of cdh15 (also 'myotubule cadherin') revealed an inverse relationship with N-cad; cdh15 expression is undetectable at 12 hpf when N-cad is broadly expressed and expressed throughout the developing muscles at 48 hpf when N-cad is down regulated (Supplementary Fig. 18a, b). This change in expression correlates tightly with the organisational transition—at 24 hpf anterior cells have replaced N-cad with Cdh15, whereas posterior cells continue to express N-cad but not cdh15, with a narrow transition zone indicating a rapid cadherin switch (Fig. 4g). Focusing on the organisational features over time revealed that muscle cells transition from a relatively unorganised, isotropic, and dense arrangement to a highly organised, anisotropic, and low-density configuration as N-cad expression decreased. This shift reflects their gradual alignment into parallel fibres, where the directional organisation of the fibres contributes to their anisotropic footprint (Fig. 4h). The N-cad negative amorphous/crystalline populations at 24 hpf exhibited almost identical feature profiles as earlier/ later time points, suggesting that the transition between archetypes was rapid. These findings are consistent with a model where N-cad expression locks cells in the amorphous archetype, an idea we tested directly by depleting N-cad function using CRISPR/Cas9 (Supplementary Fig. 19a–e). Early myotomes of N-cad knockout (KO) embryos exhibited predominantly crystalline organisation already from 12 hpf onward and throughout the observed time window (Fig. 4i, j). N-cad deficient cells maintained a high degree of anisotropic organisation,

yet exhibit reduced density throughout (Fig. 4k). The striking link between N-cad expression and the amorphous archetype raises the question if misexpression of N-cad alone is sufficient to drive an archetype transition. To test this idea experimentally, we misexpressed a N-cad-GFP fusion protein, that has previously been shown to complement function in N-cad-deficient embryos[37], in basal cells of the epidermis (Supplementary Fig. 20). While N-cad-GFP was specifically enriched at the interfaces of expressing basal cells, indicating that it mediates homotypic cell-cell interactions also in this ectopic context (Supplementary Fig. 20a–e), it did not detectably shift these or their neighbouring cells from their normal epidermal organisation (Supplementary Fig. 20f, g). This result is consistent with previous experiments showing that cadherin misexpression does not alter tissue organisation in other organisms[39]. Combined, these data show that dynamic N-cad expression is required, but unlikely sufficient, for the amorphous archetype during development.

### N-cad loss leads to amorphous-to-crystalline archetype-switching

The finding that downregulation of N-cad expression enables tissues to transition from an amorphous to a more crystalline organisation predicts that the continued high-level expression of N-cad by tissues such as the CNS could be required to maintain their amorphous archetype. To test this hypothesis, we expanded our analysis of N-cad KO to whole embryos between 12 and 48 hpf, starting at a developmental stage when N-cad has just reached its full expression (Supplementary Fig. 21a). 12 hpf saw a significant reduction in the fraction of cells showing amorphous organisation with an increase in crystalline organisation (Fig. 5a, b). Moreover, N-cad KO embryos at this stage did not exhibit a significant reduction in total cell count over WT, indicating that the identified change in organisation is not a consequence of defects in cell proliferation or viability. However, at later stages, N-cad KO embryos showed the previously described morphological phenotypes and lower cell numbers overall (Fig. 5c, Supplementary Fig. 21b, c), meaning that changes in tissue organisation at these stages could be a secondary consequence of more global developmental defects. To study the organisational effects downstream of N-cad loss more directly, we focused on the CNS, the major N-cad expressing tissue at 48 hpf. We generated identifiable N-cad deficient clones by combining a genetic mosaic approach with transgenic reporters for N-cad expression and neural identity. Mosaic knockout (mKO) of N-cad was achieved by injecting Cas9 + sgRNAs against N-cad into a single cell at the 8-cell stage (Fig. 5d). This significantly reduced the general morphological defects observed in 1-cell stage CRISPR injections or N-cad homozygous mutant lines (Supplementary Fig. 18f). To identify N-cad deficient clones, we took advantage of the fact that most cells of the CNS at 48 hpf co-express N-cad and the neuronal reporter Neural Beta-Tubulin (NBT:dsRed) (Fig. 5e). Therefore, we identify CRISPANT cells (i.e. cells that would express N-cad normally, but have lost gene function) as cells that express NBT but not N-cad in the mKO condition (Fig. 5d, e). We found a significant reduction of such double-positive cells in mKO samples over WT (Fig. 5e), confirming the efficacy of the approach. Using whole-brain light sheet microscopy and nuQ-LOUD (Fig. 5f, Supplementary Fig. 22a, b), we quantitatively compared the organisation of N-cad deficient (NBT+/Cdh2-GFP-) and surrounding WT neurons (NBT+/Cdh2-GFP+) in mKO embryos at 48 hpf. In WT animals, analysed NBT+ cells organised amorphously, as did NBT + / Cdh2-GFP+ cells in the mKO conditions (Fig. 5g, h). By contrast, NBT +/Cdh2-GFP- neurons within the same sample were assigned at a significantly higher frequency to the crystalline archetype (Fig. 5g, h). Moreover, spatial correlation analysis revealed that N-cad deficient cells surrounded by N-cad competent cells organised amorphously, while groups of N-cad deficient cells had gained crystalline organisation (Fig. 5i), exhibiting a decrease in cell density and an increase in

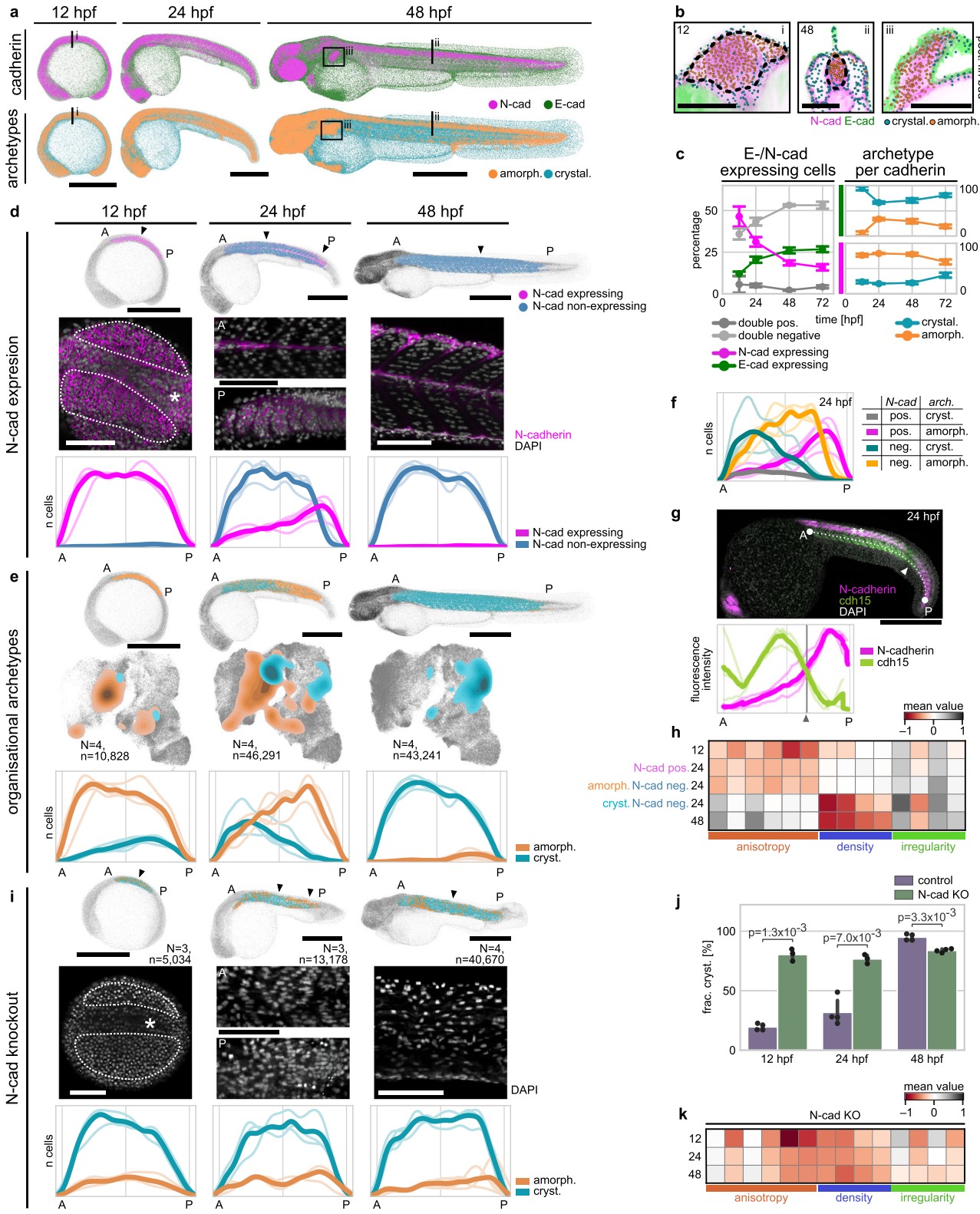

regularity (Fig. 5j), confirming that archetypes are an emergent feature of multicellular organisation. These results show that N-cad expression is not only required for the maintenance of amorphous organisation during muscle development but serves the same role during the development of the CNS. Together with the dynamic and phenotypic findings from *in toto* analysis, this supports a model where N-cad is a general driver of the amorphous archetype during zebrafish development.

## Discussion

Advances in light microscopy and transcriptomics have led to an explosion of interest in tissue heterogeneity in a wide variety of experimental systems from organoids[40] to synthetic embryos[23], and whole small organisms[17]. Progress towards a more general mechanistic understanding of tissue diversification depends on integrating data collected from different contexts[41]. The motivation behind nuQLOUD was to provide a common imaging-based framework that is flexible

**Fig. 4 | Changes in N-cad expression are linked to an amorphous-to-crystalline transition during muscle development. a**, **b** E- and N-cad expression patterns qualitatively match crystalline and amorphous archetypes. Grey: non-expressing cells, double-positive cells not shown. **c** Association of E-/N-cad with organisational archetypes persists throughout early development. $N = 4$ samples/time point. Error bars: s.d. of mean. **d** Muscles downregulate N-cad in an anteroposterior (A-P) fashion. Arrowheads: position of representative images; dotted lines: muscle progenitors at 12 hpf; asterisk: notochord. Line plots: cadherin-expressing cell numbers along A-P axis. Ordinates are not normalised between time points.
**e** Organisational archetypes switch in an A-P pattern, visualised via t-SNE and A-P line profiles. **f** Crystalline muscle cells have downregulated N-cad, while N-cad expressing cells are organised amorphously during transition. **g** N-cad down-regulation coincides with cdh15 upregulation at 24 hpf. Dashed line: measured line

profile. Arrowhead: transition point. Maximum intensity projection. ** N-cad expressing spinal cord. **h** Developing muscles become less isotropic and less dense when switching organisational archetype. **i**-**k** Transient CRISPR/Cas9-mediated N-cad KO in F1 embryos increases crystalline organisation and disrupts A-P transition at 24 hpf. **i** Representative images and A-P line profiles show loss of amorphous organisation in N-cad KO. **j** Between 12 and 48 hpf, KO significantly more crystalline cells organise crystalline than WT (Welch's $t$-test, two-sided; $N = 3$ or 4 per time point; dots: embryos; error bars: s.d. of mean). **k** N-cad-deficient cells match WT 48 hpf muscle in density but show more heterogeneous isotropy. Solid lines in A-P plots show total cell number; thin lines: individual samples. $N$ number of samples, $n$ total cells. Scale bars: embryos 500 μm, zoom-ins 100 μm. Source data are provided as a Source Data file.

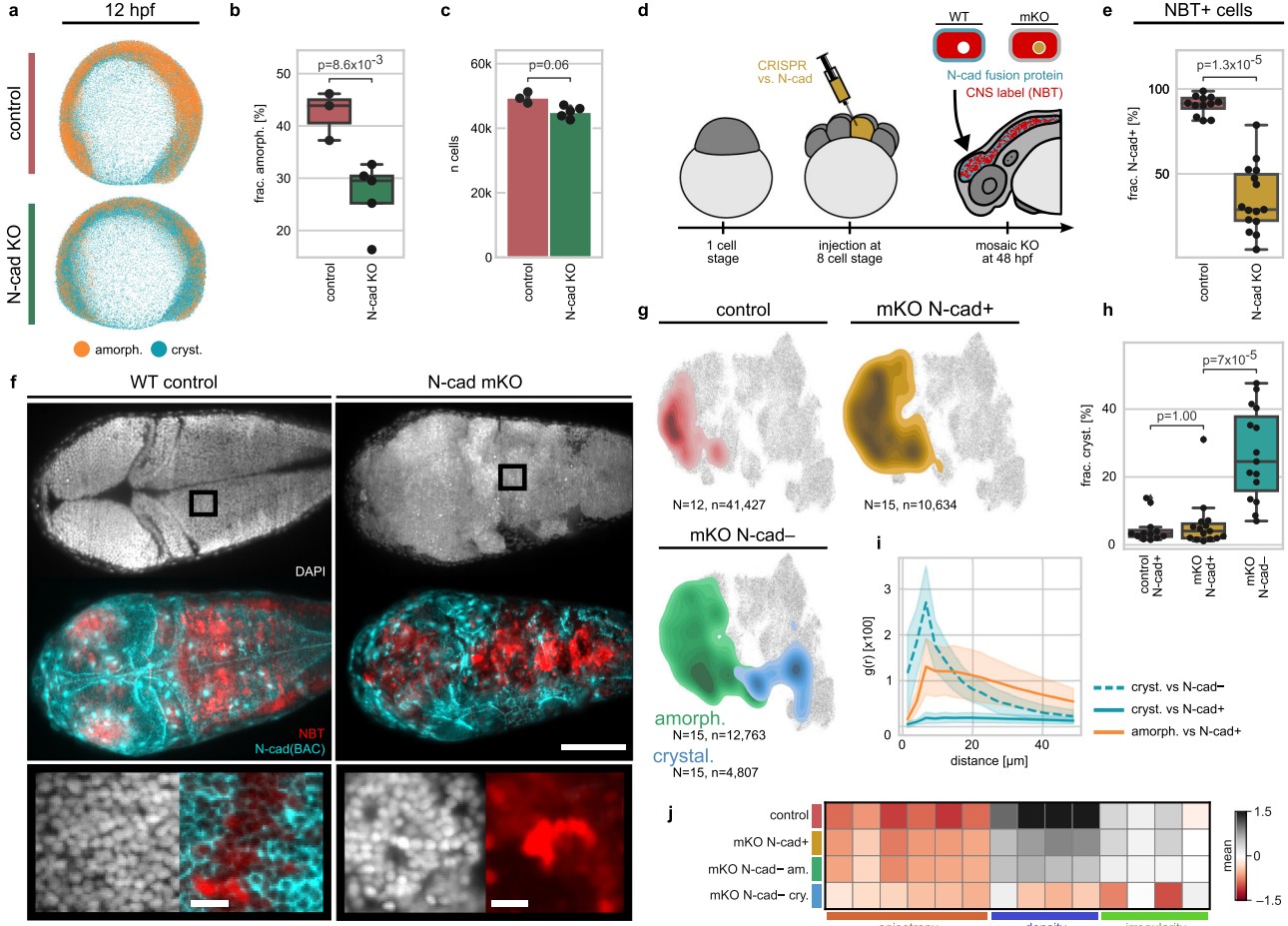

**Fig. 5 | Clonal N-cad loss induces localised shift towards crystalline archetype.**
**a**-**c** N-cad knockout (KO) reduces amorphous organisation frequency at 12 hpf.
**a** Representative renderings of control and N-cad KO embryos (nuclei segmented and coloured by archetype). KO via Cas9 + sgRNAs injection at the 1-cell stage.
**b** Fraction of amorphously organised cells drops ~30% in KO. **c** Total cell number remains unchanged between conditions. **d** Mosaic KO (mKO) of N-cad generated via Cas9 + sgRNAs injection into a single cell at the 8-cell stage. CNS cells co-express NBT and N-cad under WT conditions; mKO cells lack N-cad but express NBT, enabling identification. **e** mKO is efficient, reducing N-cad+ fraction among NBT+ cells from ~98 to ~34%. **f** Whole-brain light sheet microscopy confirms mKO phenotype and organisational changes mKO clones (overview scale bar: 200 μm; zoom-in: 15 μm; $N = 12$ control, 15 mKO). **g** NBT+N-cad- cells from mKO embryos show a

shift toward crystalline organisation in feature space, not observed in N-cad+ cells. Organisation quantified per sample and mapped onto the in toto organisational feature space ($N$: samples; $n$: total cells). **h** Crystalline fraction is significantly higher in mKO cells than N-cad+ cells, which are comparable to control. **i** N-cad-deficient cells organise crystalline only when neighbouring cells are also N-cad-deficient. Spatial correlation function g(r) shows distance-dependent likelihood of finding specific local organisation over uniform, random expectation. **j** Shift towards crystalline organisation is associated with reduced cell density and increased regularity. All p-values via Welch's $t$-test, two-sided. Dots in bar/box plots: individual embryos ($N = 3$ WT or 4 KO in (**b**, **c**); $N = 12$ WT or 15 mKO in (**e**, **h**)). Error bars/bands: s.d. of mean. Box plots show median (centre), interquartile range (box), and 1.5× IQR (whiskers). Source data are provided as a Source Data file.

and general enough to encompass the high organisational variation that is found between tissues and organisms. We achieved this goal by sacrificing cell-scale morphological features, which are more challenging to quantify and are better-suited for characterising cell-type differences, and instead focusing on how cells are arranged in 3D. This

focus on the organisation of tissues rather than cells is analogous to how architecture generally focuses on the layout of buildings rather than bricks. Reducing embryonic complexity to the positions of all cell nuclei, achieved using established in toto segmentation methods[10], allowed us to leverage the computational efficiency of point-cloud-

based frameworks and analyse large numbers of embryos each comprising >100,000 cells. We demonstrate that this can be done by simply adding DAPI, a universal DNA stain that removes the need for specific genetic reporters, thus nuQLOUD can be used to quantify tissue architecture in basically any biological specimen, essentially *gratis*.

One goal of the nuQLOUD approach was to provide a way to define tissue organisation at the whole-organism scale. General descriptors of tissue state, such as the epithelial-mesenchymal paradigm, are extremely powerful as they enable the transfer of concepts and mechanisms between a variety of biological contexts[42]. While mesenchymal and epithelial states are normally identified using diagnostic markers, they were originally defined via their distinct morphological and organisational characteristics[43]. Such architectural features, often obvious to the trained eye, are challenging to quantify and define systematically. Addressing exactly this challenge, nuQLOUD shows that embryonic complexity can be reduced to crystalline and amorphous tissue archetypes that appear analogous to epithelia and mesenchyme, respectively. However, it's clear that these tissue archetypes are not simply 'organisational proxies' of classical mesenchymal and epithelial states. For example, while the crystalline archetype comprises epithelia such as skin and pronephros, it also includes non-epithelial tissues like muscles. Likewise, the amorphous archetype includes pectoral fin mesenchyme but also branchial arches, spinal cord neurons and eyes. These tissues are amorphous, as they are denser and less positionally ordered than epithelia, but they don't fit the definition of loosely organised bona fide mesenchyme. Indeed, nuQLOUD's local organisational features select for cell collectives and tissues rather than individual mesenchymal cells. Thus, the amorphous archetype describes cells that are in between epithelia and mesenchyme on the organisational spectrum. Of relevance here is the emerging concept that such 'E/M hybrid' states, also termed partial EMT, correlate with increased fate plasticity or 'stemness' in contexts such as cancer and reprogramming[44]. Intriguingly, time-resolved analysis revealed that most internal cells in early embryos are initially amorphous before undergoing a progressive 'crystallisation' that correlates with organ progenitor assembly. Such 'amorphous to crystalline transitions' again appear analogous to mesenchymal to epithelial transitions (MET), and the related jamming transition[25,45], however they can be identified systematically from changes in cell arrangement alone. Combined, these data support the conclusion that organ maturation displays a general set of organisational features that can be reliably detected using the nuQLOUD framework, as demonstrated for example, in the maturation of fast twitch muscles.

The quantitative readout of tissue organisation provided by nuQLOUD allowed us to explore the relationship between cadherin expression and organisational archetypes. Integrating spatial expression data of 12 major cadherins into the nuQLOUD framework revealed two distinct groups: cadherins linked to crystalline tissues, such as E-cad, and those associated with amorphous tissues, including N-cad. Time-resolved analysis revealed that N-cad expression is tightly correlated with the amorphous archetype, being expressed by the majority of cells at early stages and becoming replaced by more tissue-specific cadherins as cells undergo amorphous to crystalline transitions. By contrast, cells of the nervous system displayed high N-cad expression and amorphous archetype organisation across all stages studied, consistent with its described role in nervous development. N-cad depletion, either through normal developmental downregulation or targeted perturbations, led to cells switching their organisation from amorphous to crystalline, identifying N-cad as a central regulator of the amorphous archetype. These data support a model where N-cad downregulation and exit from the amorphous archetype represents a common gating mechanism, regulating non-neural tissue assembly. Interestingly, recent findings show that inactivation of N-cad function increases the self-organisation potential of

cultured mammalian gastruloids[46], consistent with the proposal that regulation of N-cad expression may play a similar gating function during the formation of synthetic embryos. It will be very interesting to explore whether archetypal organisational patterns and transitions are also a predictive feature tissue assembly in such synthetic embryo systems. Looking ahead, we can imagine several ways in which nuQLOUD may be combined with other approaches to address key questions in tissue biology. Of general interest is how changes in the cell arrangement feedback on signalling pathways that control cell fate, whether through pathways regulated via cell-cell coupling, such as Hippo and Notch, or by changing the reach and distribution of morphogens. Recent work on organ development[47], tissue patterning[48], and immune regulation[49] has highlighted that differences in tissue organisation can be as influential in cellular decision making as signalling pathways. Developing a standard language for the quantification of cellular arrangement, like the one presented here, will be instrumental for a more integrated understanding of the interplay between genetic and morphological factors that enable contextual decision-making in physiology and disease.

The current nuQLOUD framework is not without limitations. First, it assumes a continuous arrangement over a characteristic length scale, typically one nearest neighbour on the Delaunay graph. While this assumption is appropriate for analysing tissues with dimensions exceeding ~30 μm, it is less optimal for narrow structures, such as blood vessels[50], or isolated cells, such as macrophages and fibroblasts, which are sampled along with their surrounding tissue. Second, organisational features in nuQLOUD are defined as local averages, which may average out tissues that exhibit different structural properties along different axes. For example, at later stages, the retina shows regular, layered organisation in one direction, while being more amorphous in orthogonal planes. To account for this, future versions of nuQLOUD will include directional organisational features, such as directed variation of cell density. Our investigation of cadherins identified N-cad as a key player in regulating amorphous organisation, however, the lack of phenotype resulting from inactivation of others does not mean that they play no role, as cadherins are known to act redundantly.

## Methods

### Zebrafish handling
Zebrafish (*Danio rerio*) strains (Golden) were maintained, grown and bred following the standard procedures described in ref. 51. All experiments were conducted in accordance with the regulation and guidelines of the veterinary office of the University of Zürich and the Canton of Zürich, Switzerland and the European Union Directive 2010/62/EU. Embryos were staged following[29]. All embryos used in this study were younger than 72 h post-fertilisation, a developmental stage prior to sex differentiation; therefore, sex was not considered in the experimental design.

### Chemical treatment
Embryos were kept in E3 medium and were treated with 0.002% N-phenylthiourea (PTU, Sigma-Aldrich) from 24 hpf on to prevent pigmentation. For immobilisation of embryos older than 24 hpf during screening they were treated with 0.01% tricaine methanesulfonate (MS222, Sigma-Aldrich). Moreover, highly concentrated tricaine (300 mg/L) was used to euthanize embryos prior to fixation. To aid and accelerate dechorination of embryos between 24 and 48 hpf, they were treated with 0.05% pronase[52].

### Zebrafish lines
To visualise gene expression live and without staining, transgenic zebrafish lines were used. The nervous system was visualised using Tg(NBT:DsRed)[53], E-cad expression was visualised using either TgCRISPR(cdh1-mLanYFP) or TgCRISPR(cdh1-tdTomato)[36]. N-cad

expression was visualised using TgBAC(cdh2:cdh2-GFP)[37]. Over-expression of N-cad in the skin was achieved using TgBAC(p63:Gal4)[54], TgCRISPR(cldni:cldni-mScarlett) crossed with Tg(UAS:cdh2-mNG). The N-cadherin mutant *pac*[tm101b] (cdh2[tm101/+]) was used to validate the morphological phenotype of the N-cad CRISPR[55].

## Fixation of samples and nuclear staining

Prior to imaging, zebrafish embryos at stages before 48 hpf were manually dechorionated using Dumont #5 forceps; embryos at later time points were already hatched. Embryos were euthanized using tricaine and fixed for 1 h at room temperature in 4% paraformaldehyde (PFA) in 1× phosphate-buffered saline (PBS). Subsequently, the samples were rinsed three times with PBS Tween (PBS + 0.05% Tween-20, Thermo Fisher Scientific, PBS-T), permeabilized for 1 h in PBS Triton (PBS + 0.1% Triton X-100, Thermo Fisher Scientific) and rinsed again with PBS-T. The nuclei of the samples were stained by treating the sample with 1 × 4',6-diamidino-2-phenylindole (DAPI) for 2 h at room temperature. After staining, the samples were rinsed with PBS-T, stored in PBS-T at 4 °C and imaged within 14 days.

## Fluorescent labelling of transcripts via HCR RNA FISH

To identify cells that expressed genes of interest, we labelled their transcripts using HCR RNA fluorescent in situ hybridisation (HCR RNA FISH)[56]. Probes were designed by and ordered from Molecular Instruments. Fluorescent amplifiers with emission wavelengths of 594 nm (red) or 647 nm (infra-red) were used. Generally, two sets of HCR probes were multiplexed within individual samples to increase imaging throughput. To stain samples with HCR RNA FISH, embryos were fixed for 1 h using 4% PFA. Subsequently, samples were washed for three cycles with PBS-T and permeabilised via a methanol dehydration sequence (25, 50, 75% methanol in PBS-T for 15 min, 100% methanol for 1 h). A reversed sequence was used to rehydrate samples with 10 min per step and four concluding wash cycles. Afterwards, samples were pre-hybridised using the probe hybridisation buffer at 37 °C for 30 min and HCR probes were applied (2 pmol per probe) for 12–16 h at 37 °C. To remove unbound probes, samples were incubated four times for 15 min in probe wash buffer and two times 5 min in 5× saline-sodium citrate + tween (ssct) buffer. Embryos were pre-amplified in amplification buffer for 30 min and then incubated with hairpin solution for 12–16 h. Hairpin solution was generated by mixing separately heat-shocked (30 s at 95 °C) amplifier-specific hairpin h1 and hairpin h2 solutions in amplification buffer. The staining was completed by a final five cycles of PBS-T washes. Samples were kept in PBS-T afterwards and either further processed or imaged.

## F0 CRISPR knockout and validation

F0 KO were generated via injection of a sgRNA/Cas9 mix into early-stage zebrafish embryos. sgRNAs sequences are summarised in (Supplementary Table 2). The injection mix was composed of Cas9-2xNLS protein (1 mg/μL), the sgRNAs (128 ng/μL) and phenol red (0.025%). Full KO were achieved by injecting 1–2 nL of injection mix into the yolk at the 1-cell stage of embryonic development, mKO were achieved by injecting into one cell at the 8-cell stage of development (-1 h 10 min post fertilisation at 25 °C). The Cas9 protein was sourced from the Protein Expression and Purification Core Facility of the European Molecular Biology Laboratory (EMBL) and was kept in media containing Hepes, KCl, and Glycerol at −80 °C.

To verify the efficacy of sgRNAs, the homology regions of the used sgRNAs were amplified and sequenced to detect DNA sequence polymorphisms. For that, injected embryos were grown to 4 dpf and euthanised using tricane. Genomic DNA was extracted from individual samples using 40 μL QuickExtract DNA extraction solution (Lucigen QE0905T). Embryos were incubated for 15 min at 25 °C, 5 min at 65 °C and 2 min at 95 °C and vortexed in between incubation steps. Subsequently, solid debris was removed by centrifuging 1 min at 13,000 × g.

To amplify the region of sgRNA homology, flanking primer pairs were designed using the software tool Primer3[57] with default parameters. Primer sequences are summarised in Supplementary Table 3. Polymerase chain reaction (PCR) was performed using Taq polymerase (NEB M0267S) and PCR products were analysed via gel electrophoresis. Band locations were predicted based on the size of the cut or uncut genomic DNA sequence and DNA was recovered via gel extraction and purification (QIAGEN 28604). Sanger sequencing was used to detect sequence polymorphisms which indicated sgRNA efficacy (Supplementary Figs. 15 and 19).

## Light sheet microscopy

Samples were mounted in 1% low-melting agarose solution in 1X E3 fish embryo medium by aspirating them head-first into 20 μL glass capillaries (BRAND Transferpettor caps, Merck) using Transferpettor piston rods (BRAND, Merck).

Light sheet microscopy was performed on a Zeiss Z.1 Lightsheet microscope. All images were acquired using a Zeiss W Plan Apochromat 20 × 1.0 corrected water immersion objective and a set of two Zeiss 10× illumination objectives. For *in toto* acquisition the detection objective was de-zoomed by a factor 0.45, 0.65 for partial brain imaging. The beam waist of the light sheet was optimised for maximum uniformity across the field of view. Images were recorded using scientific Complementary Metal–Oxide–Semiconductor (sCMOS) cameras (PCO edge 5.5) with 1920 × 1920 pixels. During acquisition, the camera sensors were cropped in a portrait fashion (1200 − 1600 × 1920 pixels) to exclude non-signal areas, improving acquisition speed and decreasing data volume.

For whole-organism acquisition, samples were imaged from four orthogonal sides starting with a lateral view with the ventral-dorsal axis of the sample going from left to right and continuing in steps of 90° clockwise along the anteroposterior axis of the sample. For 12 hpf samples, one such imaging volume was sufficient to capture a whole sample. For more developed samples, several overlapping (10–20%) four-view imaging volumes were acquired along the anteroposterior axis of the sample; 24 hpf: 2 volumes, 48 hpf: 4 volumes, 72 hpf: 5 volumes. The z-spacing of consecutive frames was 1 μ and for each frame the left and right light sheet illumination was recorded separately. Acquisition is illustrated in Supplementary Fig. 1. During acquisition, the 'Pivot Scan' option was turned on to reduce illumination artefacts. The light sheet offset was manually calibrated for each sample individually by contrast maximisation. Illumination settings were chosen to minimise acquisition time. To avoid photobleaching during setup, spatial alignment and multiview setup was performed using the DAPI channel. Other fluorescent channels were set up using single-frame acquisition. Multicolour imaging was performed sequentially by prioritising z-stacks over colour, i.e. full z-stacks were acquired repeatedly with single colours. The whole imaging process can hence be summarised as: light sheet direction → z-stack → colour → volume → tiles.

## Multi-view fusion of light sheet microscopy data

Raw light sheet images were stored as czi (Zeiss) files. The code repository MVRegFus (github.com/m-albert/MVRegFus), later refactored into the package multiview-stitcher[58], was used to fuse the acquired data into one image volume with isotropic resolution and locally optimised image quality. First, the opposing light sheet illuminations were fused per frame maximising the normalised discrete cosine transform Shannon Entropy (DCTS) to select the highest contrast elements for the fused image[59]. Subsequently, overlapping image volumes were fused using DCTS to select the highest contrast regions. Views were registered in a pairwise fashion, matching overlapping volumes of consecutive rotational angles and positions along the anteroposterior axis. Finally, all views were fused performing Lucy-Richardson based multi-view deconvolution, additionally considering DCTS derived multiplicative weights for each view.

For processing, voxel intensities were background subtracted considering a constant value of 200 AU. Images were binned by the factors 4 × 4 × 1 (x, y, z) during processing to increase performance. The final isotropic resolution of the fused image was 1 μm/voxel. The processed images were stored in the HDF5-derived ims format for efficient compression and handling. The fusion pipeline is illustrated in Supplementary Fig. 1.

## Nuclear segmentation via TGMM

Individual nuclei were segmented in 3D from volumetric microscopy images using the Tracking with Gaussian Mixture Models (TGMM 2.0) software[10]. The background of individual images was estimated by measuring the maximum intensity of regions not containing nuclear signal using the image analysis software Fiji[60]. The anisotropy parameter was set to 1.0 since the processed images had isotropic resolution. The minimal and maximal size of detected nuclei was determined by measuring nuclear sizes in fluorescence images and was set to 150 and 4000 voxels, respectively. To suppress mis-segmentations, the covariance matrix of the 3D Gaussian distribution was regularised such that its eigenvalues were bounded between 0.02 and 0.1, limiting the size of the three principal axis of the distribution. Additionally, total eccentricity was limited to $\epsilon = 9$. All other parameters were either left at default values or they were empirically determined to their used values. All parameters are summarised in Supplementary Table 1. The TGMM software was implemented in C++ and was released with a command line-based interface or a rudimentary graphical user interface. Both were not suitable for automated segmentation of multiple images in parallel and generally struggled with processing more than a single image at a time. We therefore developed a python-based API which managed the tool's parameters and the image- and segmentation file paths. Moreover, it enabled the parallel processing of multiple images simultaneously and automatically transformed the TGMM output files into a DataFrame representation. The software is publicly available at https://github.com/max-brambach/tgmm_utility.

## Generation of Voronoi diagrams

Three-dimensional Voronoi diagrams were generated using a modified version of the software voro++[61]. We modified the software to enable the generation of a radially restricted Voronoi diagram using command-line input, which enabled the integration of the tool into a Python-based workflow. The radial restriction was achieved by initialising every Voronoi cell as a dodecahedron with edge length 100 voxels (≜ μm). Overlapping Voronoi cells were subsequently cropped to a regular Voronoi diagram. Boundary cells were limited to their initial shape and could therefore be identified by consisting of faces that were not shared with other Voronoi cells. The modified voro++ version is publicly available at https://github.com/max-brambach/voro. To suppress boundary effects on the Voronoi diagram, we developed a method to restrict the size and shape of boundary cells based on the distribution of their neighbouring cells using auxiliary points and a secondary Voronoi diagram. See Supplementary Note 1 for more details.

## Intensity quantification

For light sheet images, fluorescence intensity was quantified following two strategies. Nuclear signals were evaluated in a 2× dilated region around the location of the segmented nucleus (Supplementary Fig. 7b). All other signal patterns such a cytosolic and more punctaed signals commonly associated with HCR RNA FISH were integrated in each nucleus' Voronoi cell (Supplementary Fig. 7c). To delete noise contributions, the signal distribution was then fitted with two Gaussian distributions. The lower mean Gaussian was considered noise, while the higher one was considered the true signal.

Fluorescence intensity line profiles used to quantify N-cad and cdh15 expression were drawn on the ventral somites of a sum intensity projection for each sample individually. N-cad and cdh15 expression was then evaluated along this line. Background noise was measured outside the sample and subtracted, autofluorescence of the sample was measured at non-fluorescing parts and used to normalise the line intensity.

## Organisational features

To characterise local tissue architecture, we extracted organisational features from the 3D coordinates of segmented nuclei. Multi-scale cell density was measured using kernel density estimation within concentric spherical regions (0–10 μm, 10–20 μm, 20–30 μm). Counts were normalised by shell volume to obtain volumetric densities. This approach distinguished compact local clusters from broader tissue-level density. We further employed restricted 3D Voronoi diagrams to quantify spatial relationships between cells. For each Voronoi cell, we computed: (i) Voronoi cell volume, reflecting local crowding and anisotropy; (ii) Voronoi density, defined as the mean inverse distance to Delaunay-connected neighbours; (iii) number of neighbours, i.e. the number of adjacent Voronoi cells sharing a face; and (iv) centroid offset, the distance between the Voronoi cell's centroid and its seed point, capturing local asymmetry. To assess neighbourhood heterogeneity, we calculated the mean and standard deviation of Voronoi volume, neighbour number, and centroid offset across each cell's immediate Voronoi neighbourhood (excluding the cell itself). This added six features describing local variability. All features were jointly z-score normalised across samples to enable cross-feature comparisons, embedding, and clustering, while preserving inter-sample differences. See Supplementary Note 2 for more details. To enhance interpretability, we grouped organisational features into three classes—density, anisotropy, and irregularity—based on their similarity using hierarchical agglomerative clustering with Ward linkage based on their pairwise Pearson correlation (Supplementary Fig. 4e, f). Correlation was assessed on a single cell level and then averaged across all samples. Features in the density class included multi-scale cell densities (0–10 μm, 10–20 μm, 20–30 μm) and Voronoi density. These features are directly related to the local packing of cells, measuring either kernel-based densities within concentric spherical shells or pairwise proximity within the Delaunay graph. The anisotropy class comprised Voronoi cell volume (mean and neighbourhood statistics) and centroid offset (individual and neighbourhood statistics). These features quantify geometric asymmetries in local cell arrangements. Voronoi volume is sensitive to anisotropic neighbourhoods, decreasing when density is concentrated along a single axis, while centroid offset measures the spatial displacement between a cell's position and the centre of its associated Voronoi polyhedron, increasing with local asymmetry. The irregularity class included the number of neighbours (individual and neighbourhood values) and the standard deviation of Voronoi cell volume in the neighbourhood. These features reflect heterogeneity in the local microenvironment. A higher number of neighbours or greater variability in Voronoi volumes indicates disordered or non-uniform cell packing, independent of absolute density. Individual feature importance was assessed by analysing how much each feature contributed to the first two principal components of the data, validating that the feature set was not over-defined (Supplementary Fig. 4g).

## Identification of organisational motifs and archetypes

Organisational motifs were identified by clustering all cells based on their 14 z-scored organisational features using a Gaussian mixture model with 11 components, initialised with k-means clustering. The number of components was empirically found using the elbow criterion on the Silhouette score and the Bayesian information criterion as well as by identifying a local minimum of the Jensen-Shannon divergence gradient (Supplementary Fig. 8a–c). The importance of each feature to this classification was evaluated by training a random forrest classifier on the feature vectors and motif labels; utilisation of

individual features gives an estimate for importance. No feature was found to be underutilised (Supplementary Fig. 8d).

Organisational archetypes were identified by clustering motifs based on their pairwise feature correlation using hierarchical agglomerative clustering and Ward linkage. All features were shown to contribute to the classification either using a random forest classifier or via feature dropout (Supplementary Fig. 11a, b). Moreover, the assignment to archetypes was not sensitive on the prior assignment to organisational motifs; initial clustering into 2, 4, 8, …, 256 clusters yielded similar results (Supplementary Fig. 11c–e), however, with decreased computational performance for higher numbers.

### Embedding of partial imaging data into *in toto* organisational feature space

Zoomed-in analysis of specific organs, such as the brain (Fig. 5) and the skin (Supplementary Fig. 16) using nuQLOUD was performed in the following way. Nuclei were segmented using TGMM and organisational features were calculated as described above. Nuclei on the border of the imaging frame were excluded from analysis due to their artificially altered feature profile. New data was normalised using the z-score parameters used to normalise the *in toto* dataset. To add new data to the existing t-SNE, we followed[62]. In brief, new points were initialised in the embedding at the coordinates of their nearest neighbours in feature space. The embedding was then run for 500 iterations at exaggeration 3 and momentum 8.

### Spinning disc confocal microscopy and corresponding data analysis

Data on cdh15 expression during muscle development and N-cad misexpression in the skin, spinning disc confocal microscopy was used. Samples were mounted in 1% low-melting agarose solution on their side in 1× E3 fish embryo medium on a Cellvis 4-Chamber Glass Bottom Dish on their side.

Confocal images were acquired at a Andor Dragonfly 200 spinning-disc microscope using a Nikon ×20/NA 0.95 water immersion objective, 1 µm z-step, spinning disc with a 40 µm pin-hole size and captured by a Sona sCMOS camera at 2048 × 2048 pixels. Tiled images were captured with 20% overlap.

Tiled images were stitched using the Grid/Collection stitching plugin[60] (default parameters, fusion method: maximum intensity). To achieve isotropic spacing, the images were then down-sampled in xy using bilinear interpolation (factor 0.35).

CDH15 expression data was processed using the nuQLOUD pipeline outlined above; N-cad misexpression data was processed in the following way. Images of the p63:Gal4, UAS:cdh2-mNG and clndi:clndi-mScarlett channels were maximum projected in z and manually adjusted to the same intensity levels using FIJI. Line profiles were exported using the 'Plot Profile' function. For plotting purposes, the intensities were normalised following:

$$I_{norm} = \frac{I - I_{min}}{I_{max} - I_{min}}, \tag{1}$$

where $I_{min}$ and $I_{max}$ denote the minimum and maximum intensities along the given line profile, respectively.

For segmenting skin cell areas, the inverted projected clndi:clndi-mScarlett channel images were imported in cellpose2[16] and segmented using the cyto2 model (cell diameter: 60px, flow_threshold: −20, cell-prob_threshold: −2, stitch-threshold: 4). Segmented objects that corresponded to the centre of lateral line organs rather than skin cells were manually removed in cellpose. Cellpose segmentations were reimported in FIJI. To minimise distortion effects from projections on the side of the fish, only the areas and mean N-cad intensities in cells in the centre of the fish were measured. For each sample, N-cad high, and N-cad low expressing cells were classified using Otsu's method. Cell areas from four samples were combined to perform a two-sided Mann–Whitney–Wilcoxon test comparing the cell areas of N-cad high and low expressing cells.

### Single cell RNA sequencing data processing

Single cell RNA sequencing data of early zebrafish development was obtained from[33] and processed using scanpy. Pre-processing was performed following[63]. Initially, cells were filtered out that expressed less than 200 genes and genes were removed that were expressed in less than 3 cells. The numbers of counts per cell were normalised to 10,000 and logarithmized. Highly variable genes were identified following[64] with minimum mean 0.125, maximum mean 3, and minimum dispersion 0.5. Individual genes were scaled to unit variance and maximum standard deviation was limited to 10. Differential gene expression analysis was performed according to[65] using the Mann–Whitney U-test.

### Statistics and reproducibility

Used statistical tests are indicated in the corresponding figure legends. If not stated otherwise, $N$ denotes the number of samples (replicates) and $n$ the number of cells (observations). For statistical analysis, observations are grouped by sample.

Box plots illustrate data distributions by showing their median (central line), interquartile range (box), 1.5× interquartile range (whiskers), and outliers (flyers). Error bars on bar plots indicate the standard deviation of the mean if not stated otherwise.

No statistical method was used to predetermine sample size. Individual sample groups contained at least three samples. Data were excluded only in cases where image quality was insufficient to permit reliable segmentation. Exclusion decisions were based on manual inspection of the raw microscopy data prior to analysis and were independent of experimental conditions. The experiments were not randomised and the investigators were not blinded to allocation during experiments and outcome assessment.

For low-dimensional embedding, either t-SNE or UMAP were used. t-SNE representations were generated following[62] using principal component initialisation, dual perplexities (50 and 500), learning rate $n/12$ and the cosine distance. 25,000 randomly selected observations were pre-embedded using 250 iterations at exaggeration 12, and momentum 0.5, followed by 750 iterations at exaggeration 1, and momentum 0.8. The remaining dataset was mapped onto the pre-embedding using a $k$-nearest neighbour search in feature space. Finally, the full embedding was optimised with perplexity 30, 500 early exaggeration iterations (exaggeration 4, momentum 0.5) and 500 iterations at exaggeration 3, and momentum 8. UMAP embeddings were generated using minimum distance 0.5, spread 1.0, alpha 1.0, gamma 1.0, negative sample rate 5 and initial spectral embedding. Spatial correlation functions were computed to quantify cell clustering. For each sample, a three-dimensional k-d tree was constructed from cell coordinates, and pairwise distances were used to estimate the local density of neighbouring cells. The pair correlation function $g(r)$ was calculated as the observed density of cell pairs within distance $r$, normalised by the expected density under a homogeneous Poisson distribution. Pair correlation functions were computed separately for N-cad negative vs negative and negative vs positive, stratified by archetype status.

### Reporting summary

Further information on research design is available in the Nature Portfolio Reporting Summary linked to this article.

## Data availability

All data supporting the findings of this study are available within this article, its supplementary files and the following online repositories. The raw light sheet data (post multi-view fusion) generated in this study have been deposited in the BioImage Archive, accession number

S-BIAD1405[66]. The scRNAseq data by Farnsworth et al.[33] used in this study are available at NCBI SRA under accession code PRJNA564810 [https://www.ncbi.nlm.nih.gov/bioproject/564810]. The scRNAseq data by Wagner et al.[67] used in this study are available at NCBI GEO under accession number GSE112294. Source data are provided with this paper.

## Code availability

A Python implementation of nuQLOUD is available at [https://github.com/max-brambach/nuQLOUD], archived under [https://doi.org/10.5281/zenodo.15733475][68]. A modified version of voro++ accompanying nuQLOUD is available at [https://github.com/max-brambach/voro]. Tools to efficiently run nuclear segmentation using TGMM are available at https://github.com/max-brambach/tgmm_utility, archived under [https://doi.org/10.5281/zenodo.15733494][69]. The multi-view fusion algorithm MVRegFus is available at [https://github.com/m-albert/MVRegFus], archived at [https://doi.org/10.5281/zenodo.15240470][58].

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

## Acknowledgements
Francesca Peri and lab, François Nédélec, Damian Brunner, and all members of the Gilmour lab for discussion. Cornelia Henkel for fish care. Urs Ziegler, Jana Döhner, Joana Delgado Martins of the UZH Center for Microscopy and Image Analysis for support with and infrastructure for image acquisition and analysis. Kim Remans and the Protein Expression and Purification Core Facility of the European Molecular Biology Laboratory for Cas9. Mark Cronan and David Tobin for E-cad reporter lines. SNSF Grant 310030_204834 to D.G. and UZH Forschungskredit FK-21-084 to M.B.

## Author contributions
M.B. and D.G. conceived the project and designed experiments. MB performed all experiments, performed light sheet microscopy, analysed images, developed the nuQLOUD software, and performed data analysis. JW performed N-cad misexpression experiments and data analysis. J.W. performed analysis of cdh15 expression in muscles. MA developed multi-view fusion software. M.B. and J.J. performed F0 CRISPR KOs. RB provided transgenic zebrafish lines and optimised HCR staining. M.B. and D.G. interpreted the data and wrote the manuscript with input from all authors.

## Competing interests
The authors declare no competing interests.
