## [Transparent Peer Review file · Nature Communications]

In toto analysis of embryonic organisation reduces tissue diversity to two archetypes requiring specific cadherins

Corresponding Author: Professor Darren Gilmour

Version 0:

Reviewer comments:

Reviewer #1

(Remarks to the Author)

The manuscript "In toto analysis of multicellular arrangement reduces embryonic tissue diversity to two archetypes that require specific cadherin expression" by Brambach et al. presents a new image-analysis tool that allows the description of 3D structures by determining the positions of nuclei relative to each other. The method is tested on three developmental stages of zebrafish embryogenesis. The authors found a correlation between the observed configurations of nuclei ("archetypes") and the expression of cadherin molecules. The functional relevance of the correlations is then tested by knockout experiments that show that the function of specific cadherins is required for the observed patterns. Overall, the authors present a novel tool for characterizing embryonic tissue architecture that nicely complements other approaches, in particular single-cell chromatin and transcriptome analyses among others. I believe the paper is of interest to the readership of the journal and provide below a few points that the authors could address to further improve the paper.

More important issues-

1. An important point is that the authors do not validate the accuracy of their segmentation. For example, based on the magnified view in fig S1c it is not clear that the segmentation is indeed robust. This point should be examined for tissues the authors specifically refer to. Also, manipulations in cadherins could in principle affect nuclear staining and thus affect the detection of the nuclei. For example, the precision of nuclei identification could differ between fig5f bottom left and right (nuclei magnification) and affect the conclusions.
2. The authors use the terms "amorphous" and "crystalline" and link them to the field of material science. While such arrangements are consequential concerning the physical properties of the material, this point is not examined here. It would be better would be to define the states differently, or state that this relates to the organization in the tissue and not connect it to material science.
3. In "while cadherins are known to act redundantly, loss of the.....". Cadherin 2 does not show significance in figure S12d-f.
4. Contrary to the statement in the text, grey-labelled N-cad positive crystalline cells are observed in fig.4f.
5. It is not clear how, according to the definitions, there can be an overlap between the amorphous and crystalline domains in fig 4e and 4fb (T-sne plots).

Minor points

6. In several positions of the article, the authors use language that could perhaps change the language. The findings presented in the paper are interesting and would have a strong impact on the field, so there is no need to mention 'not seeing the forest for trees', "tissue biology dates to Aristotle" etc
7. The method the authors report on complements other approaches used in the field rather than being superior or better than other procedures. Thus, there is no need to list the shortcomings of different methods, in particular since some of the statements are not accurate. e.g. Cell morphology segmentation methods encourage a focus on cell-scale features that may not be relevant to architecture at the tissue-scale, which can lead to 'not seeing the forest for trees'. , "contrast" their method with previous ones – just say that it is complementary.". Cell morphology segmentation is very relevant if not more informative, than following the organization of nuclei.
8. The effect of the arrangement of the nuclei on tissue properties (point 4) could be examined by following the evolution of the patterns in time (not just 3 distinct stages) to observe how transitions occur. Such experiments would provide insights

into the relationships between the arrangements of nuclei and tissue properties (fluidity etc). Documenting the actual transition would probably be beyond the scope of this paper.

9. Discuss the possible relevance of the tissue architecture as measured/defined here for the evolution of actual tissue shape and function.

10. "Comparing the hierarchy of transcriptional and organisational similarity revealed that amorphously organised cadherin domains included more transcriptionally similar cells, while crystalline organised cadherin domains were more distinct in their gene expression profile, highlighting that diverse sets of gene expression may result in similar organisational patterns." – what is this statement based on? fig 3f?

11. To make it more clear, in fig 4d, use a different color for the N-cad-, since it is used for E-cad in fig 4c. Best would be not to define it as N Cad – and +, but define them as "NCad expressing cells" and "cells not expressing NCad". Same in S6 etc.

12. It is not clear what information movies 1-3 provide regarding the "anterior-posterior N-cad distribution."

13. Improving the color choice would help in several cases – e.g. fig 1g and 4f (The grey cluster is not clearly visible).

14. "Early myotomes of N-cad knockout (KO) embryos exhibited predominantly crystalline organisation already from 12 hpf onward and throughout the observed time window (Fig. 4 e,f)", and "N-cad deficient cells maintained a high degree of anisotropic organisation, yet exhibit reduced density throughout (Fig. 4 h,g)." citation error - fig4 h,i and fig4h, j should be cited respectively.

15. In S15b,c add "(hpf)" to the x-axis and add significance value.

16. There are multiple positions where spelling/spaces/duplications should be corrected. e.g. in

"To enable the investigation of the organisational diversity of entire developing organisms, we initially focussed on zebrafish embryos at 48 hours post fertilisation (hpf), a stage where the the progenitors of several major organs are already formed²⁵."

"One example for a maintained motif was motif IX, which consistently mapped to the YSL from 12 hpf."

"[...] these analyses demonstrate how nuQLOUD can be used to highlight connections between transcriptional and tissue organisational patterns, links that can then be tested with perturbations using the same quantitative framework."

"Focusing on the organisational features over time revealed that muscle cells transition from an relatively unorganised, isotropic, and dense arrangement to a highly organised, anisotropic, and low-density configuration as N-cad expression decreased."

"This significantly reduced the general morphological defects observed in 1-cell stage CRISPR injections or N-cad homozygous mutants lines (Suppl. Fig. 14 f)."

These data support a model where N-cad downregulation and exit from the amorphous archetype represents a common gating mechanism regulating non-neural tissue assembly.

"While cadherins are known to act redundantly³⁰, [...] resulted in their target tissues shifting detectably shifting towards the crystalline archetype (Suppl. Fig. 12 d-f)."

"Focusing on the organisational features over time revealed that muscle cells transition from an relatively unorganised [...]"

"[...] indicating that the identified change in organisation isn't a consequence of defects in cell proliferation or viability." "This significantly reduced the general morphological defects observed in 1-cell stage CRISPR injections or N-cad homozygous mutants lines"

(Remarks on code availability)

Reviewer #2

(Remarks to the Author)

Advances in single cell RNA-seq technologies have facilitated the characterization of cellular diversity in multicellular organisms. However, the organization of such diverse cell types into functional tissues is still poorly understood due to the lack of methods to classify tissue architectures. In this work, the authors use in toto imaging and point cloud modelling to classify different tissue domains in the developing zebrafish embryo. The workflow they have developed is independent of cell or tissue types and uses only nuclear signal, thus allowing for widespread applications.

The study shows that the entire embryo can be divided into two principal tissue archetypes – 'amorphous' and 'crystalline'. The two archetypes also correlate with the expression of specific cadherin transcripts. It further shows that among these cadherins, N-cadherin is required for maintaining the amorphous state during zebrafish axis elongation.

The study is well written and contains a number of potentially very interesting findings. However, while the method for classifying tissue organization and correlating them to the expression of specific cadherins is impressive, the biological significance of the 'amorphous' and 'crystalline' archetypes is less clear. I have the following major concerns, followed by a few minor suggestions:

1. The tissue organization observed here develops from 12 hours to 48 hours post fertilization. During these stages, the tissue architecture is also influenced by interaction of cells with the extracellular matrix (ECM). How does the two archetypes correlate with the expression of ECM components? Does ECM influence the organization of the tissues into the two archetypes?

2. Figures 4 and 5 suggest that loss of N-cadherin leads to switch of the tissue archetype from 'amorphous' to 'crystalline'. Specifically, during the development of twitch muscles, the transition from 'amorphous' to 'crystalline' happens due to decrease in irregularity but anisotropy and density remains the same. On the other hand, mKO N-cad leads to reduced density and irregularity in the brain. How to interpret these different results? Are the two transitions the same or tissues have multiple way to achieve the transitions? And how does this transition relate to the function of the tissue?

3. In mKO of N-cadherin, ~25% of mutant cells change from 'amorphous' to 'crystalline' archetype. Are there certain cells that are more susceptible to this transition in the absence of N-cadherin? What could be the other factors that contribute to tissue archetype?

4. The transition from 'amorphous' to 'crystalline' archetype is correlated with loss of N-cadherin expression. Does this also correlate with changes in the expression of other cadherins, particularly that observed in 'crystalline' tissues? Would it be possible to change the tissue archetype by cadherin overexpression?

5. What are the factors that drive the 'crystalline' organization of tissues? From figure S12 f it appears that *cdh1* KO also results in more cells becoming 'crystalline'. Does this mean that tissues become 'crystalline' in the cadherin mutants because they lose cell adhesion, resulting in reduced density and irregularity? It could be helpful to more carefully describe the *cdh1* KO phenotype in the results section.

Minor suggestions –

- Figure 4j is not described or references in the results section
- Supp Fig 12 e is denoted as Supp Fig 12 d in the figure legend.
- In the figure legend for Supp Fig 5b, does the red regions denotes markers expression levels or cadherin levels?

(Remarks on code availability)

-

Reviewer #3

(Remarks to the Author)

Review of NCOMMS-24-76857-T

Title: In toto analysis of multicellular arrangement reduces embryonic tissue diversity to two archetypes that require specific cadherin expression

Authors: Max Brambach, Marvin Albert, Jerome Julmi, Robert Bill, Darren Gilmour

Brambach et al. introduce nuQLOUD, a computational framework for analyzing tissue architecture by reducing cellular arrangements to 3D nuclear point clouds. Applying this to zebrafish embryos, the authors identify two distinct tissue archetypes (amorphous and crystalline). They demonstrate a strong correlation between these archetypes and cadherin expression patterns, with N-Cadherin playing a pivotal role in maintaining amorphous organization. Through genetic perturbation and time-resolved analysis, the study establishes N-Cadherin as a key driver of transitions between tissue states during embryogenesis. This framework provides a systematic approach to quantifying tissue organization and exploring molecular drivers of architectural diversity.

This paper addresses an interesting and highly relevant question regarding the tissue structure in developing embryos. We greatly appreciate the breadth of experiments performed and the wide spectrum of data analysis steps undertaken. Additionally, the inclusion of numerous SI figures provides a comprehensive visual overview that enhances the paper's accessibility and depth.

However, there are a few aspects that could be improved. The computational methods are described in limited detail, which makes it challenging to fully assess their validity and reproducibility. Furthermore, the sequential use of multiple clustering approaches is confusing and makes it difficult to follow the rationale behind the analytical workflow. Clarifying these aspects would strengthen the paper significantly.

Major comments:

Clustering. Throughout the manuscript, clustering is used at multiple stages in the analysis pipeline. Our major concern is the limited information about how all the different clustering steps are performed and evaluated. Let's start with the feature clustering:

Please add to the Main text that HAC with Ward linkage is used as the clustering approach

How are the three feature clusters (anisotropy, density, irregularity) defined after the clustering? The density class naming makes sense since all four underlying features represent density, but what about the anisotropy and irregularity classes?

The paper would benefit from a more extensive Methods section describing the clustering approaches for each of the classifications.

Regarding the organisational clustering:

Can the author comment on the importance of the 14 features. Which features of the 14 are most important for clustering? Is the method using all 14 features, or is it overfitting on one/a subset of the features? Can you quantify the importance of every feature?

Figure 2c shows high correlation between some features. Are all 14 necessary to obtain the two archetype classes? What if some features are left out?

The Methods mentions that the neighbourhood features are local averages. Can the authors explain in more detail how the local averages are computed? What is the “radius” of averaging, and how does this affect the outcomes?

Regarding the two organisational archetypes:

In our understanding, the archetypes are obtained by clustering the eleven organisational clusters, where the organisational clusters are obtained by clustering of the features. Is this two-step process necessary? Why not simply clustering the feature-space directly into two classes? This seems possible, since the two archetypes clearly separate in the t-SNE feature embedding.

We advise the authors to be more consistent and informative with the nomenclature of the multitude of clusters used: organisational features, organisational signatures, organisational motifs, organisational archetypes, features, feature classes, etc.

Why is 48 hpf chosen as the timepoint to do the clustering?

Along the lines of comment 2a-b, can you authors show whether the amorphous/crystalline classification is based on all 14 features, or a subset one them? Would it be possible to a similar classification using a subset of features?

Visual examples of structure. The different clusters of cells and their names (anisotropic, dense, irregular, amorph, crystalline, etc.) are used throughout the manuscript. Apart from their names being seemingly random (see comments 1.b), the manuscript would benefit greatly from visual confirmations of the different cluster types. Would it be possible to show representative parts of the point clouds that demonstrate visually the difference between these types? Are the tissues classified as “crystalline” visually different from “amorph” point clouds, or are the classes merely the outcome of the clustering magic? We understand that large point clouds printed on paper may not always be clear, but it is hard to trust all qualitative claims when we cannot visually relate them to characteristics in the point clouds. It would be great if the authors could find a way to share these point clouds on the web or in some other interactive form.

Segmentation validation. All analyses and conclusions are based on the segmented nuclei, but the authors do not show any validation of the segmentation.

Show example crops of image data together with the TGMM segmentation for visual validation.

Is the segmentation quality equal between dense and sparse areas within the embryo?

Comment on how robust the nuQLOUD results are when segmentation is suboptimal

What about tuning the TGMM parameters to make the segmentation deliberately worse, and see the effect on the resulting classes/conclusions?

Reproducibility. What if someone has imaged a small part of Zebrafish tissue, can the resulting point cloud be classified into your 2 archetypes? In other words, can a new feature vector be embedded into the same t-SNE map? If yes, please provide an example script with this application.

Throughout the manuscript, the t-SNE embedding of organisational features is displayed multiple times.

Is it unclear whether the shown t-SNE embeddings in Figures 1/2/3/5 are the same. If they are the same, are they calculated using all available cells of multiple embryos at all time points?

How are the NBT+ cells (Figure 5g) projected onto the existing embedding?

These days UMAP is typically used and preferred, especially in scRNAseq studies, it could be good to have a supplementary figure comparing the t-sne with UMAP representation

Github. We appreciate the efforts of the authors to openly share the code. However, the code is segmented over multiple repositories, has limited readme's and no example data. We recommend:

The readme's could be improved, please add: dependencies (numpy, pandas, matplotlib, tqdm, vedo, scipy, scikit-learn, seaborn, openTSNE, tiffile, h5py, etc), instructions for installation, etc.

The entire nuQLOUD pipeline consists of many steps. Would it be possible to provide a small data (crop), and have a notebook that shows the entire pipeline: segmentation, Voronoi, feature extraction, archetype classification?

We are unable to test the nuQLOUD example notebook, since the “raster_geomtery” module (used in IntensityMapping.py) is unknown.

Minor comments:

The figures and videos definitely undersell the work. Much more effort should be put into making the figures a bit more clear and even more work is required for the videos to really showcase the very interesting results of the paper.

Please add Wan et al. (Cell 2019), Lange et al (Cell, 2024), Amat et al. (Nature Methods, 2014), and Moyle et al. (Nature, 2021) as citations to the statement “in toto imaging ... large-scale tracking ... during zebrafish gastrulation” in order to fairly cover the contributions in the field.

Out of interest, does the Voronoi tessellation in any way match the actual shape of the cells? Because one of the features is the number of neighbors, I am wondering whether the number of Voronoi neighbors matches the number of actual touching neighbors of the cell.

Can the authors comment on whether the expensive Voronoi tessellation is necessary? Can you imagine a set of features purely calculated from the point cloud that allows extraction of the same classes?

The authors use t-SNE and UMAP embeddings in the manuscript. Can the authors comment on why they sometimes choose UMAP over t-SNE, and why t-SNE is chosen for the vast amount of embeddings in the manuscript? Also, please comment on the stochastic nature of t-SNE (vs. PCA for example), are the results affected by the t-SNE giving different outputs when repeatedly applied to the same data?

Can the authors comment on the ignored dynamics of the cells? Are the cell movement dynamics considered slow enough that the amorphous/crystalline structure are constant? Can the authors give an estimate for the speed of cell dynamics, relative to the camera integration time?

The submission package contains 3 SI videos, however the videos are not mentioned in the manuscript. Also, volume rendering is probably more interesting than the current slice-by-slice view. This can be done rather easily using Napari or any other similar tool that supports 3D rendering

Typos/captions/figures:

“Muscles depends on their being bundled” > remove “their” (introduction)

“A stage where the the progenitors” > remove double “the” (results)

“From transgenic zebrafish lines and via hybridisation” > remove “and” (results)

“Of multiple samples.N: num” > missing space before “N” (caption Figure 1)

“Solid lean” > “solid line” (caption SI Figure 4)

The red text “segmentation” in panel a of Figure 1: technically the red dots seem to represent the cell centers, not the segmentation

Figure 1g: the caption does not contain information about what the colored lines represent

Figure 2b: there is a manuscript-wide inconsistency in whether there is a space before “hpf” (“12hpf” vs. “12 hpf”)

Figure 2de: the overlay of the two archetypes over the embryos is good as an overview, but not very informative. Can you show zooms of certain areas within the embryo that show clear examples of the amorphous/crystalline nature of the tissue?

Figure 2: the caption contains important information that is not present in the Main text. Please make sure the information is present in both (for example that HAC is used as the clustering algorithm)

Figure 3d: the colors are confusing. Do the pink/green overlays on the t-SNE plots represent the two archetypes or the “cdh2”/“cdh1”?

Figure 4: the panel indications are not horizontally aligned (for example “c”)

On multiple occasions in the manuscript, the numerical notion $\text{mean} \pm \text{std}$ has the wrong number of significant figures. For example, “116,600 \pm 9,000” should be “117e3 \pm 9e3” or “(117 \pm 9)e3”, since the std can only have 1 significant figure.

Part of the Microscopy section of the Methods has a different vertical line spacing

(Remarks on code availability)

Th repo requires a lot of work: there is no readme, the dependencies are not specified, and there is overall very little effort to make the work reproducible or even useful for others. This is unfortunate because the rest of the work and manuscript is interesting. Improving the repo and making it truly useful for others would really make a difference and improve the impact and utility of this work for others.

Version 1:

Reviewer comments:

Reviewer #1

(Remarks to the Author)

We find the responses satisfactory, consider the paper suitable for publication in Nature Communications, and congratulate the authors for the nice work.

(Remarks on code availability)

Reviewer #2

(Remarks to the Author)

he manuscript has been revised in accordance with the suggestions provided by the three reviewers. All of my comments have been thoroughly and convincingly addressed. The revised version shows significant improvement and is now, in my view, suitable for publication in Nature Communications.

(Remarks on code availability)

-

Reviewer #3

(Remarks to the Author)

I'm quite impressed with how thoroughly the authors addressed the major methodological concerns raised in the reviews. The addition of comprehensive segmentation validation (Suppl. Fig. 2) with manual annotation of >4000 nuclei and quantitative performance metrics substantially strengthens the foundation of all downstream analyses - this was a critical missing piece. The expanded visual examples, particularly the interactive 3D renderings and motif projections (Suppl. Fig. 9), now provide much needed intuitive access to what these "amorphous" vs "crystalline" classifications actually look like in real tissue architecture, which makes the biological claims much more convincing. The computational methodology is now much better explained with proper justification for the two-step clustering approach and feature importance analyses that demonstrate all 14 features contribute meaningfully. Most importantly for reproducibility, the GitHub repositories have been transformed from essentially unusable to a well-documented framework with example data and complete pipelines. The additional analyses (spatial correlation functions, ECM investigation, cadherin switching dynamics) add nice biological depth without cluttering the main narrative. While some of the nomenclature still feels a bit heavy-handed, and I still think the connection to material science properties could be more carefully stated, these are relatively minor issues now. The paper has evolved from an interesting but methodologically underdetermined study into a robust, reproducible framework that should be quite valuable for the community. The biological insights about N-cadherin's role in maintaining amorphous organization are well supported and the archetype concept provides a useful conceptual framework for thinking about tissue organization across development.

(Remarks on code availability)

Much improved!

REVIEWER COMMENTS

Reviewer #1 (Remarks to the Author):

The manuscript “In toto analysis of multicellular arrangement reduces embryonic tissue diversity to two archetypes that require specific cadherin expression” by Brambach et al. presents a new image-analysis tool that allows the description of 3D structures by determining the positions of nuclei relative to each other. The method is tested on three developmental stages of zebrafish embryogenesis. The authors found a correlation between the observed configurations of nuclei (“archetypes”) and the expression of cadherin molecules. The functional relevance of the correlations is then tested by knockout experiments that show that the function of specific cadherins is required for the observed patterns. Overall, the authors present a novel tool for characterizing embryonic tissue architecture that nicely complements other approaches, in particular single-cell chromatin and transcriptome analyses among others. I believe the paper is of interest to the readership of the journal and provide below a few points that the authors could address to further improve the paper.

Response:

We thank reviewer 1 for their supportive comments.

More important issues –

1. An important point is that the authors do not validate the accuracy of their segmentation. For example, based on the magnified view in fig S1c it is not clear that the segmentation is indeed robust. This point should be examined for tissues the authors specifically refer to. Also, manipulations in cadherins could in principle affect nuclear staining and thus affect the detection of the nuclei. For example, the precision of nuclei identification could differ between fig5f bottom left and right (nuclei magnification) and affect the conclusions.

Response 1.1:

We agree that it is helpful to validate the accuracy of the segmentation (a point also raised by reviewer 3). In the revised manuscript we have now included a detailed evaluation of the nuclear segmentation performance using three approaches (Suppl. Fig. 2).

(1) Visual inspection of overlays between nuclear images and segmentation masks did not reveal qualitative mismatches.

(2) To assess segmentation performance quantitatively, we manually annotated 4212 nuclei across 9 randomly selected fields of view from 3 independent samples of the tissues we refer to. Comparison with automated segmentation revealed a true positive rate of $97.8 \pm 0.9\%$, a false positive rate of $1.3 \pm 0.8\%$, and a false negative rate of $0.9 \pm 0.5\%$.

(3) While our segmentation proved to be sufficiently robust, nuQLOUD should nevertheless tolerate less complete segmentation as long as the errors are evenly distributed across the feature space. We confirmed that the rare erroneously segmented nuclei were not enriched in specific organisational contexts such as high-density regions, providing further assurance that potential segmentation errors have negligible impact on our findings.

Regarding the potential impact of cadherin perturbations on nuclear staining and detection, previous studies have established that DAPI, as a chromatin stain, provides reliable nuclear labelling of zebrafish embryos of different genotypes. We observed comparable numbers of nuclei across cadherin perturbation conditions; thus it is unlikely that nuclear staining is systematically affected by cadherin manipulation.

We are grateful to the reviewers 1 and 3 for pointing out this shortcoming of the original manuscript, we feel that the revised version is strengthened by data confirming that our nuclear staining and segmentation is both robust and precise across the analysed datasets.

2. The authors use the terms “amorphous” and “crystalline” and link them to the field of material science. While such arrangements are consequential concerning the physical properties of the material, this point is not examined here. It would be better would be to define the states differently, or state that this relates to the organization in the tissue and not connect it to material science.

Response 1.2:

We thank the reviewer for pointing out the potential for misinterpretation and offering possible solutions. We are happy to apply the second suggestion, to state clearly that the chosen terms relate to the organisation of the tissue rather than connecting it to material science and mechanical properties.

3. In “while cadherins are known to act redundantly, loss of the.....”. Cadherin 2 does not show significance in figure S12d-f.

Response 1.3:

We agree that Cadherin 2 does not show a significant shift in global archetype scores in Figure S12d–f. As discussed in the final results section, we believe this may be due to the severe morphological disruption caused by global Cdh2 knockout, which compromises the ability of our in toto screen to detect organisational changes. This prompted us to pursue more targeted approaches, such as tissue-specific analysis of the somites and mosaic knockouts, which reveal clear organisational phenotypes upon Cdh2 loss. We have removed the reference to Cdh2 from the original sentence to avoid confusion.

4. Contrary to the statement in the text, grey-labelled N-cad positive crystalline cells are observed in fig.4f.

Response 1.4:

We agree with the reviewer and have changed the phrasing to ‘few’. We believe that the detected N-cad positive, crystalline cells are part of the fast-twitch muscles, a superficial layer of cells at the somite boundary which continue expressing N-cadherin. We illustrate the existence of those cells in the Suppl. Movies 2-3.

5. It is not clear how, according to the definitions, there can be an overlap between the amorphous and crystalline domains in fig 4e and 4fb (T-sne plots).

Response 1.5:

The reviewer has raised an interesting point. Our interpretation is that the apparent overlap arises from two factors. First, due to the stochastic nature of the T-SNE embedding, complete separation of classes is not guaranteed. While T-SNE effectively preserves local relationships, some degree of overlap between populations is common, particularly on the periphery of classes. Nevertheless, clustering analyses performed in the full feature space confirm that the separation between amorphous and crystalline domains is robust. Second, the contour lines in the T-SNE plots are drawn independently for each class without normalisation across classes. This choice was made to better visualise small populations that would otherwise be difficult to detect. As a result, contour areas may appear slightly inflated for less frequent classes, contributing to the visual impression of overlap. We also performed dedicated analyses to ensure that the classification into organisational archetypes is an inherent property of the data and not dependent on specific choices in the clustering strategy. For example, we show that varying the number of initial sub-clusters (termed “motifs”) does not substantially impact the final clustering into archetypes (Suppl. Fig. 11 c-e), supporting the robustness of the observed separation.

Minor points

6. In several positions of the article, the authors use language that could perhaps change the language. The findings presented in the paper are interesting and would have a strong impact on the field, so there is no need to mention 'not seeing the forest for trees', "tissue biology dates to Aristotle" etc

Response 1.6:

We're pleased that the reviewer considers the findings sufficiently interesting and impactful that there's no need to resort to such phrasing. We're happy to remove the highlighted statements, especially "not seeing the forest for trees", which may be interpreted as criticism of other methods (see response 1.7). However, we'd like to keep the later "focussing on the forest rather than the trees" which we feel helps distil our coarse-grained approach to tissue organisation in an easily assimilated way.

7. The method the authors report on complements other approaches used in the field rather than being superior or better than other procedures. Thus, there is no need to list the shortcomings of different methods, in particular since some of the statements are not accurate. e.g. Cell morphology segmentation methods encourage a focus on cell-scale features that may not be relevant to architecture at the tissue-scale, which can lead to 'not seeing the forest for trees'. , "contrast" their method with previous ones – just say that it is complementary.". Cell morphology segmentation is very relevant if not more informative, than following the organization of nuclei.

Response 1.7:

We thank the reviewer for highlighting that these statements are not helpful and thank them for suggesting alternative phrasing. We agree that 'complementary' is the correct way to describe how nuQLOUD relates to other segmentation approaches.

8. The effect of the arrangement of the nuclei on tissue properties (point 4) could be examined by following the evolution of the patterns in time (not just 3 distinct stages) to observe how transitions occur. Such experiments would provide insights into the relationships between the arrangements of nuclei and tissue properties (fluidity etc). Documenting the actual transition would probably be beyond the scope of this paper.

Response 1.8:

We thank the reviewer for this suggestion and agree that it could provide additional insights into the nature of archetype transitions. However, given that the underlying mechanisms are likely to be context dependent this will require significant effort to be useful, thus we agree that it goes beyond the scope of this first study.

9. Discuss the possible relevance of the tissue architecture as measured/defined here for the evolution of actual tissue shape and function.

Response 1.9:

We thank the reviewer for this thoughtful question. This obviously goes beyond our expertise and any direct evolutionary and functional interpretations are inherently speculative. However, we believe that the emergence of two recurring organisational archetypes—amorphous and crystalline—may reflect distinct functional and developmental strategies with evolutionary relevance.

Crystalline organisation, characterised by high anisotropy and low local density, tends to correlate with terminal differentiation and tissue-level function, as seen in epithelia and muscle. In contrast, amorphous organisation is often associated with early developmental stages, plasticity, or proliferative capacity. This is consistent with the observation that many tissues transition from amorphous to crystalline organisation as they mature and acquire specialised functions. Interestingly, exceptions such as the CNS, which maintains an amorphous state despite high functional complexity, suggest that these organisational modes are not rigidly tied to function but may represent different solutions to architectural constraints.

In evolutionary terms, one could speculate that the ability to switch between these archetypes enabled multicellular organisms to balance organisational flexibility during development with structural robustness in differentiated tissues. This echoes hypotheses by Brunet and King (e.g. Brunet & King, 2018) on the evolution of multicellularity, where precise spatial organisation of diverse cell types was proposed to underpin the emergence of tissue-level function.

10. “Comparing the hierarchy of transcriptional and organisational similarity revealed that amorphously organised cadherin domains included more transcriptionally similar cells, while crystalline organised cadherin domains were more distinct in their gene expression profile, highlighting that diverse sets of gene expression may result in similar organisational patterns.” – what is this statement based on? fig 3f?

Response 1.10:

We thank the reviewer for drawing attention to this point. To answer the question, the original statement was based on the clustering analysis shown in Fig. 3 f, where we observed that amorphously organised cadherin domains tended to group more tightly at the transcriptional level, whereas crystalline domains appeared more transcriptionally heterogeneous. We recognise, however, that this observation was only briefly presented and not fully supported by additional data in the main text, which may have made the statement confusing or open to misinterpretation.

This comment encouraged us to take a closer look at the previously published single-cell RNA-seq dataset (Farnsworth et al., 2018), performing a differential gene expression analysis comparing individual cadherin-expressing domains to other cells within the same cadherin class. This revealed some interesting trends—for instance, cells within amorphously organised N-cadherin-expressing domains showed enrichment for broadly expressed genes such as *nova2* and *hmgb1b*, whereas crystalline domains expressing E-cadherin or *Cdh15* were enriched for tissue-specific genes such as *epcam* and *actc1a* (Rev. Fig. 1). These findings are consistent with our original interpretation, suggesting that similar organisational patterns can arise from more diverse gene expression states in crystalline tissues.

That said, we ultimately feel that this line of reasoning, while intriguing, detracts from the focus of the manuscript and introduces complexity without significantly advancing our central claims. We have therefore decided to remove the statement from the revised version of the manuscript. Nonetheless, we appreciate the reviewer’s question as it highlighted this issue. For completeness, we include the results of the extended gene expression analysis here (Rev. Fig. 1), in case the reviewer may find it of interest.

Reviewer Figure 1:

A: Highly differentially expressed genes for each cadherin domain versus the remaining cells of the dataset. Expression domains of the N-cadherin-like class exhibit similar differentially expressed genes such as *nova2* or *hmgbl1b*, while the differentially expressed genes are less shared between members of the E-cadherin-like class. Method: t-test with Benjamini-Hochberg correction. Y-axes indicate the z-score of the gene's expression used for the calculation of the p-value.

B: Summary of the differential expression analysis. Differentially expressed genes are co-expressed in the N-cadherin-like class (orange box), while differentially expressed genes are more specific to individual cadherin domains in the E-cadherin-like class (blue box). Grey value of the dots represents the mean expression level, size of the dots indicates the fraction of expressing cells per domain.

11. To make it more clear, in fig 4d, use a different color for the N-cad-, since it is used for E-cad in fig 4c. Best would be not to define it as N Cad – and +, but define them as “NCad expressing cells” and “cells not expressing NCad”. Same in S6 etc.

Response 1.11:

We thank reviewer Reviewer 1 for pointing this potentially confusing colour choice, *Ncad*- cells are now shown in blue to avoid overlap with *Ecad*. We also see that *Ncad* + or *Nad* – could be interpreted as genotypes, we have now consistently replaced these with expressing/non-expressing to improve clarity.

12. It is not clear what information movies 1-3 provide regarding the “anterior-posterior N-cad distribution.”

Response 1.12:

We agree that the inclusion of these movies should be better justified. The expression analysis we've described here shows a wave of N-cad suppression in the muscles. However, this is not so obvious when observing *Ncad:Ncad-GFP* lines due to a single superficial layer of muscle that continue to express GFP. The function of these Z-stack movies is to show that the described downregulation is visible also in these lines using optical sectioning.

13. Improving the color choice would help in several cases – e.g. fig 1g and 4f (The grey cluster is not clearly visible).

Response 1.13:

We have removed the embedding in Fig. 4f as it was redundant with the accompanying quantification and a source of confusion

14. “Early myotomes of N-cad knockout (KO) embryos exhibited predominantly crystalline organisation already from 12 hpf onward and throughout the observed time window (Fig. 4 e,f)”, and “N-cad deficient cells maintained a high degree of anisotropic organisation, yet exhibit reduced density throughout (Fig. 4 h,g).” citation error - fig4 h,i and fig4h, j should be cited respectively.

Response 1.14:

We thank the reviewer for pointing out this error, which has been corrected.

15. In S15b,c add “(hpf)” to the x-axis and add significance value.

Response 1.15:

We thank the reviewer for pointing out this oversight, which has been addressed

16. There are multiple positions where spelling/spaces/duplications should be corrected. e.g. in “To enable the investigation of the organisational diversity of entire developing organisms, we initially focussed on zebrafish embryos at 48 hours post fertilisation (hpf), a stage where the the progenitors of several major organs are already formed²⁵.”

“One example for a maintained motif was motif IX, which consistently mapped to the YSL from 12 hpf.”

“[...] these analyses demonstrate how nuQLOUD can be used to highlight connections between transcriptional and tissue organisational patterns, links that can then be tested with perturbations using the same quantitative framework.”

“Focusing on the organisational features over time revealed that muscle cells transition from an relatively unorganised, isotropic, and dense arrangement to a highly organised, anisotropic, and low-density configuration as N-cad expression decreased.”

“This significantly reduced the general morphological defects observed in 1-cell stage CRISPR injections or N-cad homozygous mutants lines (Suppl. Fig. 14 f).”

These data support a model where N-cad downregulation and exit from the amorphous archetype represents a common gating mechanism regulating non-neural tissue assembly.

“While cadherins are known to act redundantly³⁰, [...] resulted in their target tissues shifting detectably shifting towards the crystalline archetype (Suppl. Fig. 12 d-f).”

“Focusing on the organisational features over time revealed that muscle cells transition from an relatively unorganised [...]”

“[...] indicating that the identified change in organisation isn’t a consequence of defects in cell proliferation or viability.”

“This significantly reduced the general morphological defects observed in 1-cell stage CRISPR injections or N-cad homozygous mutants lines”

Response 1.16:

We are grateful to the reviewer for taking the time to highlight these mistakes (and apologise that this was necessary). We have corrected them all in the revised version.

Reviewer #2 (Remarks to the Author):

Advances in single cell RNA-seq technologies have facilitated the characterization of cellular diversity in multicellular organisms. However, the organization of such diverse cell types into

functional tissues is still poorly understood due to the lack of methods to classify tissue architectures. In this work, the authors use in toto imaging and point cloud modelling to classify different tissue domains in the developing zebrafish embryo. The workflow they have developed is independent of cell or tissue types and uses only nuclear signal, thus allowing for widespread applications.

The study shows that the entire embryo can be divided into two principal tissue archetypes – ‘amorphous’ and ‘crystalline’. The two archetypes also correlate with the expression of specific cadherin transcripts. It further shows that among these cadherins, N-cadherin is required for maintaining the amorphous state during zebrafish axis elongation.

The study is well written and contains a number of potentially very interesting findings. However, while the method for classifying tissue organization and correlating them to the expression of specific cadherins is impressive, the biological significance of the ‘amorphous’ and ‘crystalline’ archetypes is less clear. I have the following major concerns, followed by a few minor suggestions:

Response:

We thank reviewer 2 for their constructive input.

1. The tissue organization observed here develops from 12 hours to 48 hours post fertilization. During these stages, the tissue architecture is also influenced by interaction of cells with the extracellular matrix (ECM). How does the two archetypes correlate with the expression of ECM components? Does ECM influence the organization of the tissues into the two archetypes?

Response 2.1:

We agree with the reviewer that how ECM influences tissue architecture is an interesting additional question. Providing an answer during revision is obviously challenging, especially as it's known that finding specific tissue shaping roles for ECM in vivo isn't straightforward, due to genetic pleiotropy and redundancy (see fibronectin deficient mice by the Hynes lab as an example of pleiotropy, SPARC deficient mice generated by DG's PhD as an example of redundancy). However, recent work defining roles for collagens in shaping specific tissues (eg feather follicles by Shyer and Rodrigues, fly CNS by Stramer) gave us hope that exploring this new avenue may provide something of additional interest for aficionados of this field.

As the reviewer refers to the impact of ECM more generally, we began analysing the entire ‘matrisome’ of 904 ECM proteins and regulators (Nauroy et al. 2018). We used cadherin expression as a reliable proxy for organisational archetypes, based on the findings presented in Fig. 3, reasoning that any association between ECM components and archetypes should emerge from co-expression profiles. As shown in Suppl. Fig. 15, this analysis did not reveal any consistent patterns linking specific ECM components with either archetype. Instead, we observed heterogeneous co-expression of matrisome genes across cadherin-expressing populations.

In a complementary spatial approach, we performed HCR RNA-FISH analysis of collagens, as candidate ECM components that could correlate with tissue archetypes and analysed their local organisation of their expressing tissues using the nuQLOUD framework. We selected the most common fibrillar collagen (colla2), col6a1 and the definitive basement membrane collagen col4a. These proved to be expressed by both epidermal keratinocytes and mesenchymal fibroblasts (Rev. Fig. 2), establishing immediately that collagen expression spans both archetypes. As mentioned in the manuscript, as nuQLOUD analysis is based on local organisational features, scattered amorphous cells like fibroblasts become blended with surrounding tissues, shifting the analysis towards a crystalline archetype, which explains why col1a2 and col6a, both expressed broadly by amorphous fibroblasts, associate with the crystalline archetype. Nevertheless, col4a6-expressing cells displayed only a ~60% likelihood of being crystalline (Rev. Fig. 3).

In summary, while we don't disagree with the reviewer's suggestion that ECM may be expected to influence organisational archetypes, analysing the transcriptomic co-expression of the entire matrisome and performing HCR and nuQLOUD analysis of candidate collagens, did not identify compelling candidates for archetype-specific regulators. While perhaps disappointing, this result was arguably predictable as ECM proteins have rarely been identified as specific regulators of tissue-shapes in previous large-scale genetic screens. These unclear findings are in stark contrast to those we present for cadherins, which can be unequivocally assigned to archetypes using either approach.

Reviewer Figure 2: Col1a2 mRNA expression @48 hpf. A. Max projection. A'. Zoom focussed on expression in epidermal basal cells and fibroblasts of the horizontal and vertical myosepta (tenocytes). B'. Single z-slice focussed on notochord associated fibroblasts. C'. Single z-slice focussed on epidermal basal cells.

Reviewer Figure 3: nuQLOUD analysis of collagens col1a2, col4a6, col6a1. a. Max projection of collagen mRNA expression at 48 hpf. b. Localisation of collagen expressing cells in t-SNE embedding of organisational feature space. N: number of samples, n: number of cells. c. Average organisational feature profiles of collagen expression domains. d. Fraction of crystalline organising cells per collagen expression domain. e. Fraction of

crystalline organising cells for cells expressing either single collagens, pairwise combinations or all three. f. Corresponding feature profiles.

2. Figures 4 and 5 suggest that loss of N-cadherin leads to switch of the tissue archetype from 'amorphous' to 'crystalline'. Specifically, during the development of twitch muscles, the transition from 'amorphous' to 'crystalline' happens due to decrease in irregularity but anisotropy and density remains the same. On the other hand, mKO N-cad leads to reduced density and irregularity in the brain. How to interpret these different results? Are the two transitions the same or tissues have multiple way to achieve the transitions? And how does this transition relate to the function of the tissue?

Response 2.2:

We thank the reviewer for raising this point and subsequent questions. We do not see the mentioned mismatch between the two transitions. In both contexts — the natural transition during muscle development following N-cad downregulation and the forced transition in the brain following N-cad inactivation — we observe a consistent reduction in density and irregularity, which are the two core features underlying the archetypal switch from amorphous to crystalline. These shared features suggest that N-cadherin loss leads to a general shift toward crystalline organisation, regardless of tissue type. The reviewer is correct that anisotropy increases during muscle differentiation; this reflects the alignment of cells into elongated fibres. This feature is not seen in the N-cadherin loss-of-function conditions, likely because in those cases, the tissue never acquires the capacity to organise into such fibre-like structures in the first place. In other words, N-cadherin is required before many of these anisotropic arrangements can be established. We have clarified this distinction in the revised text.

The purpose of our study is to understand the emergence of tissue architectural diversity, a large topic in its own right. How this transition relates to tissue function is another question, albeit interesting, and one that will have many context-dependent answers. We can assume the increased order and anisotropy associated with the crystalline archetype generally facilitates tissue functions (eg the function of most epithelial organs depends on cells being ordered and polarised). We briefly touch on this in the context of muscle development, where the emergence of crystalline organisation coincides with the formation of aligned contractile fibres—presumably to generate force in a coordinated fashion. Conversely, in the brain, amorphous organisation may allow dense packing of neuronal cell bodies in a spatially constrained environment. These functional interpretations are obviously speculative, and we would rather limit them in the manuscript, however they are consistent with the structural changes we observe.

3. In mKO of N-cadherin, ~25% of mutant cells change from 'amorphous' to 'crystalline' archetype. Are there certain cells that are more susceptible to this transition in the absence of N-cadherin? What could be the other factors that contribute to tissue archetype?

Response 2.3:

We thank the reviewer for raising this issue, the basis of phenotypic heterogeneity is an interesting general topic. The finding that only ~25% of N-cadherin deficient cells transition from an 'amorphous' to a 'crystalline' archetype is likely be attributed to three main factors:

- (1) **Genetic Expressivity:** Cells of identical genotype can vary in the extent to which they express an expected phenotype (hence 'expressivity'). While phenotypic heterogeneity is a feature that comes baked into any genetic study—there exist many null mutants where only a fraction of cells shows the phenotype—the mechanisms underlying expressivity and penetrance are rarely understood. This is especially true in complex non-isogenic contexts (e.g. fish embryos) where any number of background modifiers can impact on phenotypic strength.
- (2) **Collective nature of nuQLOUD analysis:** As noted in the Discussion, nuQLOUD quantifies local organisation based on neighbourhood-averaged features, limiting its resolution at small scales. This means that the classification of a given cell depends on its surrounding context rather than its intrinsic properties. Isolated cells or small clusters may not be identifiable as a distinct organisational state.

- (3) **Collective nature of tissue organisation:** Tissue emerges at the level of cell populations rather than single cells. In this context, N-cadherin deficient cells embedded within predominantly wild-type populations remain classified as amorphous, as their local environment retains that structure. Only when N-cadherin deficient cells form sufficiently large clusters do they exhibit crystalline organisation.

To test specifically factor (3), we calculated the spatial correlation function $g(r)$ (see updated Methods) for N-cadherin-negative cells classified as crystalline or amorphous, vs. N-cadherin-positive and negative cells. The results confirm that N-cadherin deficient cells organise crystalline when surrounded by other N-cadherin deficient cells but remain amorphous when surrounded by N-cadherin-competent ones. Thus, we could identify cells that are more susceptible to the transition, providing an answer the reviewer's first question, and offer a plausible reason for their increased susceptibility. Moreover, these data support the general concept that local organisation is strongly influenced by neighbourhood identity, consistent with the principles outlined above. We're grateful to the reviewer for prompting this analysis as we feel it led to an interesting addition to the study (Fig. 5 j).

4. The transition from 'amorphous' to 'crystalline' archetype is correlated with loss of N-cadherin expression. Does this also correlate with changes in the expression of other cadherins, particularly that observed in 'crystalline' tissues? Would it be possible to change the tissue archetype by cadherin overexpression?

Response 2.4:

In their first question, the reviewer astutely points out that our classification of cadherins into amorphous and crystalline groups offers an opportunity to extend the concept of 'cadherin switching', to date described as the transition from E-cad to N-cad expression during EMT. We're pleased to add data observing a tight inverse relationship between N-cad and Cdh15 expression. At 12 hpf, N-cadherin is expressed throughout the developing muscles while Cdh15 is absent but by 48 hpf it's the reverse (Suppl. Fig. 17). This change in expression correlates with the organisational transition – at 24 hpf anterior cells have replaced N-cad with Cdh15, whereas posterior cells continue express N-cad, with a tight transition zone that indicating a rapid switch (Fig. 4 g). Therefore, we confirm the reviewer's suggestion that the transition from N-cad+/amorphous to N-cad-/crystalline organisation coincides with a switch from N-cad to Cdh15, a cadherin of the crystalline class and thus extend the concept of cadherin switching. Interestingly, we could see no evidence that genetic inactivation of E-cad or N-cad leads to concomitant upregulation of cadherins of the opposing class – N-cad does not become upregulated in E-cad deficient skin clones (Suppl. Fig. 19 g) and E-cad does not become upregulated in N-cad deficient neuronal clones (Suppl. Fig. 22 c, d) – supporting the interpretation that cadherin switching is unlikely to be a direct consequence of cadherin loss alone and is rather coupled to other gene regulatory processes.

We are also happy to provide experimental data that directly addresses their second question, that is, whether cadherin overexpression (ie. misexpression) is sufficient to change the tissue archetype. To do this, we misexpressed N-cad in the developing epidermis, an example of the crystalline archetype tissue (Suppl. Fig. 20). A key step in any such experiments is to first ensure that the misexpressed proteins are functional. Here we used a Ncad-GFP construct that we had previously shown can perfectly complement N-cad in loss of function mutants, a gold standard measure of functionality (Revenu et al). When strongly expressed in the epidermis, Ncad-GFP was specifically enriched at interfaces with other Ncad-GFP expressing cells, consistent with it mediating homotypic cell-cell interactions, further confirmation that the construct remains functional when expressed ectopically. However, high level Ncad-GFP misexpression did not lead to detectable changes in cellular morphology or tissue organisation, suggesting that cadherin expression alone is insufficient to reprogramme tissue architecture in this context. Such a result is perhaps expected given previous work showing that cadherin misexpression is insufficient to change tissue organisation in *Drosophila* embryos (Schäfer et al. 2014 doi:10.1242/jcs.139485).

We have incorporated both findings into the revised manuscript and thank the reviewer for encouraging this useful extension of our study.

5. What are the factors that drive the 'crystalline' organization of tissues? From figure S12 f it appears that *cdh1* KO also results in more cells becoming 'crystalline'. Does this mean that tissues become 'crystalline' in the cadherin mutants because they lose cell adhesion, resulting in reduced density and irregularity? It could be helpful to more carefully describe the *cdh1* KO phenotype in the results section.

Response 2.5:

We thank the reviewer for this thoughtful comment, which we have addressed through further analysis. While Figure S12f (now Suppl. Fig. 14 f) may suggest a trend toward a crystalline shift in the E-cad knockout (*cdh1* KO), we note that this shift is not statistically significant. Given the overall strong morphological disruption observed in E-cad knockouts (as shown in Suppl. Fig. 14 c), we reasoned that a more targeted analysis would provide clearer insights.

Following the strategy used in Figure 5, we performed a mosaic E-cad knockout and focussed on cells of the basal cell layer of the embryonic skin that were labelled using constructs driven under p63 promoter. In wild-type conditions, p63-expressing cells robustly co-express E-cad. In contrast, in E-cad mosaic knockout condition, 27% of p63-positive cells lost E-cadherin expression. Importantly, among p63-positive cells, we observed a small but statistically significant decrease in the proportion of cells adopting crystalline organisation in E-cad negative cells compared to positive. We have expanded the description of the *cdh1* KO phenotype in the revised Results section and Suppl. Fig. 19 to reflect this additional analysis, and we thank the reviewer for prompting this improvement.

Minor suggestions –

- a. Figure 4j is not described or references in the results section
- b. Supp Fig 12 e is denoted as Supp Fig 12 d in the figure legend.
- c. In the figure legend for Supp Fig 5b, does the red regions denotes markers expression levels or cadherin levels?

Response 2.6:

We thank the reviewer for pointing out these errors, which have all been addressed in the revised version.

Reviewer #2 (Remarks on code availability):

-

Reviewer #3 (Remarks to the Author):

Review of NCOMMS-24-76857-T

Title: In toto analysis of multicellular arrangement reduces embryonic tissue diversity to two archetypes that require specific cadherin expression

Authors: Max Brambach, Marvin Albert, Jerome Julmi, Robert Bill, Darren Gilmour

Brambach et al. introduce nuQLOUD, a computational framework for analyzing tissue architecture by reducing cellular arrangements to 3D nuclear point clouds. Applying this to zebrafish embryos, the authors identify two distinct tissue archetypes (amorphous and crystalline). They demonstrate a strong correlation between these archetypes and cadherin expression patterns, with N-Cadherin playing a pivotal role in maintaining amorphous organization. Through genetic perturbation and time-resolved analysis, the study establishes N-Cadherin as a key driver of transitions between

tissue states during embryogenesis. This framework provides a systematic approach to quantifying tissue organization and exploring molecular drivers of architectural diversity. This paper addresses an interesting and highly relevant question regarding the tissue structure in developing embryos. We greatly appreciate the breadth of experiments performed and the wide spectrum of data analysis steps undertaken.

Response:

We appreciate reviewer 3's encouraging comments.

Additionally, the inclusion of numerous SI figures provides a comprehensive visual overview that enhances the paper's accessibility and depth. However, there are a few aspects that could be improved. The computational methods are described in limited detail, which makes it challenging to fully assess their validity and reproducibility. Furthermore, the sequential use of multiple clustering approaches is confusing and makes it difficult to follow the rationale behind the analytical workflow. Clarifying these aspects would strengthen the paper significantly.

Major comments:

Clustering. Throughout the manuscript, clustering is used at multiple stages in the analysis pipeline. Our major concern is the limited information about how all the different clustering steps are performed and evaluated. Let's start with the feature clustering: Please add to the Main text that HAC with Ward linkage is used as the clustering approach. How are the three feature clusters (anisotropy, density, irregularity) defined after the clustering? The density class naming makes sense since all four underlying features represent density, but what about the anisotropy and irregularity classes?

The paper would benefit from a more extensive Methods section describing the clustering approaches for each of the classifications.

Response 3.1:

We thank the reviewer for these thoughtful suggestions for improvement. The three feature classes—density, anisotropy, and irregularity—were defined post hoc based on hierarchical clustering of the z-scored organisational features across all cells and samples. The density class grouped together features that directly reflect local cell packing, including kernel-based density estimates at multiple spatial scales and Voronoi density, as noted by the reviewer. For the anisotropy class, we included Voronoi cell volume (mean and neighbourhood statistics) and centroid offset (individual and neighbourhood statistics), as both sets of features quantify spatial asymmetry in local cellular arrangements. Voronoi volume is sensitive to directional bias in cell placement—i.e., it is reduced in elongated or skewed neighbourhoods—and centroid offset increases as the Voronoi cell becomes geometrically asymmetric around its seed point. The irregularity class captures local heterogeneity in neighbourhood topology and geometry. It includes the number of neighbours (individual and neighbourhood statistics) and the standard deviation of Voronoi volume in the neighbourhood. These features quantify disorder in local cell arrangement: for example, a high number of neighbours or high variability in Voronoi volumes reflects irregular packing that deviates from uniform lattice-like structures. We have revised the text in the Methods section to clarify this classification rationale more explicitly and to include more detail on the clustering approaches for the classifications. Moreover, we have included the suggested details about the clustering in the main text.

Regarding the organisational clustering:

Can the author comment on the importance of the 14 features. Which features of the 14 are most important for clustering? Is the method using all 14 features, or is it overfitting on one/a subset of the features? Can you quantify the importance of every feature?

Response 3.2:

The 14 organisational features were designed to comprehensively capture distinct aspects of 3D nuclear arrangement, including density, anisotropy, regularity, and local heterogeneity. While some features may be partially correlated, our analyses suggest that each provides unique and complementary information.

To quantify their contributions, we used two complementary approaches:

1. **Principal component analysis (PCA)** on the full feature set revealed that all features—except for the standard deviation of Voronoi-based density in the local neighbourhood—contributed comparably to the variance captured by the first two principal components, which together explain the major axes of variation in the dataset. This suggests that the feature space is well-distributed and not dominated by a small subset of features.
2. **Random forest feature importance** was used to assess the influence of each feature on clustering outcomes. We trained a classifier on 100,000 randomly selected nuclei using organisational motif or archetype labels. This analysis showed that features across all categories contributed to classification, with anisotropy and density features being the most informative for distinguishing archetypes. Importantly, no single feature was sufficient to explain the classification alone, and excluding individual features consistently reduced performance (as described in our feature-dropout analysis, see below).

Together, these results confirm that the method does not overfit to a narrow subset of the feature space. Instead, the classification into organisational motifs and archetypes draws on the full diversity of spatial features, reflecting the complex and multidimensional nature of tissue organisation.

We include these analyses in the Supplementary Information (Suppl. Fig. 4 g, Suppl. Fig. 8 d) and briefly discuss them in the main text of the revised submission.

Figure 2c shows high correlation between some features. Are all 14 necessary to obtain the two archetype classes? What if some features are left out?

Response 3.3:

We appreciate the reviewer's interest in the contribution of individual features. To clarify, Figure 2c does not display correlations between individual features but rather shows the similarity of organisational motifs based on their full feature profiles. Feature–feature correlations are shown in Suppl. Fig. 4 e, where some moderate correlations are observed, but none exceed ~50%. As discussed in response to a previous comment, our analyses indicate that all 14 features contribute to defining local organisation. Principal component analysis showed that all but one feature contributes similarly to the overall variance in the dataset (as represented by the first two principal components), suggesting minimal redundancy (Suppl. Fig. 4 g). Additionally, our clustering and classification approaches are robust to feature collinearity.

To explicitly test the impact of leaving out features, we performed a feature-dropout analysis: omitting each feature one at a time, re-clustering the data into two archetypes, and comparing the resulting classification to the original using the Adjusted Rand Index (ARI). The ARI quantifies the similarity between two clusters, with a value of 1 indicating perfect agreement and 0 corresponding to random assignment. In all cases, the ARI dropped relative to the full feature set, confirming that each feature provides information important for archetype assignment. While some features may individually contribute less than others, omitting them reduces the resolution and consistency of the classification. We have included this feature dropout analysis in the revised Supplementary Information (Suppl. Fig. 11 b).

The Methods mentions that the neighbourhood features are local averages. Can the authors explain in more detail how the local averages are computed? What is the “radius” of averaging, and how does this affect the outcomes?

Response 3.4:

We are happy to elaborate on this aspect. The neighbourhood features were computed using the topological neighbourhood defined by the restricted 3D Voronoi diagram. Specifically, for each

segmented nucleus, we identified all adjacent Voronoi cells—i.e., those sharing a face in the Voronoi tessellation, which corresponds to direct connections on the Delaunay graph. This adjacency defines a discrete and biologically meaningful local neighbourhood without requiring a fixed spatial radius. For each such neighbourhood, we computed the mean and standard deviation of three primary features: Voronoi cell volume, number of neighbours, and centroid offset. Importantly, we excluded the central cell from this computation to avoid conflating the cell-intrinsic value with the neighbourhood average. This approach avoids imposing an arbitrary radius and instead adapts to local cell density and arrangement, which we found to be more robust across tissues with varying cell spacing. We have clarified this point in the Methods section.

Regarding the two organisational archetypes:

In our understanding, the archetypes are obtained by clustering the eleven organisational clusters, where the organisational clusters are obtained by clustering of the features. Is this two-step process necessary? Why not simply clustering the feature-space directly into two classes? This seems possible, since the two archetypes clearly separate in the t-SNE feature embedding.

Response 3.5:

We agree that clustering the feature space directly into two classes is in principle possible and would also recover the two archetypes. Indeed, we find that the classification into archetypes is largely conserved when varying the number of initial clusters (2, 4, 8, 16, 32, 64, 128, 256), as shown by accuracy and F1 score comparisons (Suppl. Fig. 11 c-e). This supports the idea that the archetypes are an inherent feature of the data and do not depend on the intermediate number of motifs.

We opted for a two-step approach for both interpretability and practical reasons. First, the intermediate motif clusters offer a useful representation of local structural modes that facilitates biological interpretation. Second, directly clustering all ~4 million cells using hierarchical methods would be computationally prohibitive, as algorithms like agglomerative clustering do not scale well with large datasets. While other scalable clustering approaches could be used, we chose to emphasize the modular structure of the data to guide the narrative flow. We have clarified this rationale in the revised text.

We advise the authors to be more consistent and informative with the nomenclature of the multitude of clusters used: organisational features, organisational signatures, organisational motifs, organisational archetypes, features, feature classes, etc.

Response 3.6:

We thank the reviewer for this comment and agree that our manuscript introduces a number of terms to describe different layers of analysis. We made a conscious effort to define each term precisely and use them consistently throughout the text. For example, “organisational features” refers to the quantitative measurements extracted from the spatial arrangement of cells; “organisational motifs” refers to patterns identified through unsupervised clustering of these features; and “archetypes” refer to the two dominant and recurring clusters across tissues.

While we revisited the terminology during revision and streamlined where possible, we found it difficult to replace these terms with alternatives that would be both more accurate without introducing new ambiguities. We obviously remain open to any explicit replacement suggestions, but we feel that the current nomenclature reflects the underlying concepts and is used consistently.

We hope that the clarifications added in the revised manuscript and supplementary figures help improve the clarity of the framework for the reader.

Why is 48 hpf chosen as the timepoint to do the clustering?

Response 3.7:

We chose 48 hpf for clustering because this stage provides a sweet spot of high ‘contrast’ tissue organization in an embryo with dimensions that are small enough to facilitate efficient in toto imaging. While recent studies have demonstrated the power of applying in toto imaging to

understand earlier developmental events (eg Lange et al doi: 10.1016/j.cell.2024.09.047), our aim here was to sample a wide range of different tissue organisations. By 48hpf the progenitors of many recognizable organs – skin, muscles, CNS, pectoral fins – are more clearly distinguishable than they are at earlier stages. Later stages, such as 72 hpf, do not add much in terms of organizational diversity while having the disadvantage of greater dimensions, which elongate acquisition time and generate larger data sets that are more unwieldy.

Along the lines of comment 2a-b, can you authors show whether the amorphous/crystalline classification is based on all 14 features, or a subset one them? Would it be possible to a similar classification using a subset of features?

Response 3.8:

We addressed this question using a feature-dropout approach (Suppl. Fig. 11 b), where we systematically removed each feature and assessed how well the resulting clustering preserved the original archetype classification. This analysis showed that no single feature alone drives the classification, and that all 14 features contribute to distinguishing between the amorphous and crystalline archetypes. While a subset of features can partially recover the classification, the clearest and most robust separation is achieved when the full feature set is used. These results confirm that the archetypes reflect an integrated property of cellular arrangement rather than a single structural metric.

We added this analysis to the Supplementary Information and briefly discuss it in the main text.

Visual examples of structure. The different clusters of cells and their names (anisotropic, dense, irregular, amorph, crystalline, etc.) are used throughout the manuscript. Apart from their names being seemingly random (see comments 1.b), the manuscript would benefit greatly from visual confirmations of the different cluster types. Would it be possible to show representative parts of the point clouds that demonstrate visually the difference between these types? Are the tissues classified as “crystalline” visually different from “amorph” point clouds, or are the classes merely the outcome of the clustering magic? We understand that large point clouds printed on paper may not always be clear, but it is hard to trust all qualitative claims when we cannot visually relate them to characteristics in the point clouds. It would be great if the authors could find a way to share these point clouds on the web or in some other interactive form.

Response 3.9:

We appreciate the reviewer’s interest in gaining more intuitive access to the structures underlying our classifications. To address this, we have added visual representations of both the organisational motifs and the organisational archetypes:

1. Organisational motifs (Suppl. Fig. 9): We now show maximum intensity projections of 100 μm spherical neighbourhoods centred on a representative nucleus for each motif (i.e., the one closest to the motif’s average feature profile). These provide a qualitative sense of local structure and help relate feature combinations to spatial patterns.

2. Organisational archetypes (Supplementary Media): We now provide 3D point cloud renderings of segmented embryos at 12, 24, and 48 hpf, colour-coded by archetype. These are viewable in any standard web browser and allow for interactive inspection of tissue architecture and archetype distribution. We agree that such interactive visualisation is the most effective way to evaluate qualitative claims about large 3D datasets, and we are happy to make these data openly accessible.

We have also expanded the Methods section to more clearly explain the rationale behind the terminology used to describe motifs and archetypes. While we respectfully disagree with the characterisation of the terms as “seemingly random,” we hope the expanded explanations and visual examples clarify the connections between feature space and biological interpretation.

Segmentation validation. All analyses and conclusions are based on the segmented nuclei, but the authors do not show any validation of the segmentation. Show example crops of image data together with the TGMM segmentation for visual validation. Is the segmentation quality equal between dense and sparse areas within the embryo? Comment on how robust the nuQLOUD results are when segmentation is suboptimal

What about tuning the TGMM parameters to make the segmentation deliberately worse, and see the effect on the resulting classes/conclusions?

Response 3.10:

We thank the reviewer (and reviewer 1) for raising this important point. We have now included a detailed segmentation validation analysis in the revised Supplementary Information.

Specifically:

1. Visual validation: We provide example crops of raw nuclear images overlaid with the corresponding TGMM segmentation masks for visual inspection (Supplementary Fig. 2). These confirm qualitative agreement across both dense and sparse regions of the embryo.

2. Quantitative benchmarking: We manually annotated 4212 nuclei across 9 randomly selected fields of view from 3 embryos and compared them to the automated segmentation. This revealed a high true positive rate ($97.8 \pm 0.9\%$), with low false positive ($1.3 \pm 0.8\%$) and false negative ($0.9 \pm 0.5\%$) rates, demonstrating overall high segmentation accuracy.

3. Error distribution: We assessed the feature space positions of erroneously segmented nuclei and found no enrichment in specific regions, indicating that segmentation errors are evenly distributed and do not bias the identification of organisational motifs or archetypes.

Regarding the suggestion to deliberately degrade segmentation quality by altering TGMM parameters: while technically feasible, we believe this exercise would not provide meaningful insights. We would expect that lowering segmentation quality leads to noisier feature estimation and reduced classification sharpness. However, given that nuQLOUD is designed to work on well-segmented nuclear point clouds, we do not advocate processing poorly segmented data further. Instead, we provide clear metrics and visual examples to help researchers assess segmentation quality before applying the method.

We hope this analysis adequately addresses the concern and supports the robustness of our conclusions.

Reproducibility. What if someone has imaged a small part of Zebrafish tissue, can the resulting point cloud be classified into your 2 archetypes? In other words, can a new feature vector be embedded into the same t-SNE map? If yes, please provide an example script with this application.

Response 3.11:

Yes, this is possible and indeed already implemented in our analysis of N-cadherin mosaic knockouts (Fig. 5), although we realise this may not have been made sufficiently clear in the manuscript and have now added comments to clarify this point.

Briefly, to project new data into the established feature space, we z-score the new feature vectors using the same normalisation parameters derived from our reference dataset (the ~4 million nuclei). This ensures compatibility with the original feature space without altering its structure. It is essential that the input coordinates are in micrometres and appropriately corrected for imaging anisotropy (e.g. from confocal z-stacks).

To visualise new data in the t-SNE embedding, we initialise each new point at the location of its nearest neighbour in the original feature space (found via k-d tree search) and then run the embedding for a small number of epochs to allow local adjustment without changing the global structure. However, we would like to emphasise that all analyses are performed in the full 14-dimensional feature space; the t-SNE embedding serves only for visualisation and is not used for quantitative classification.

We do not feel that is necessary to provide an example script, as this process is straightforward and widely used — it involves standard normalisation and nearest-neighbour embedding, which can be readily implemented using existing packages.

We now provide more details on embedding of partial and new data in the method section.

Throughout the manuscript, the t-SNE embedding of organisational features is displayed multiple

times.

Is it unclear whether the shown t-SNE embeddings in Figures 1/2/3/5 are the same. If they are the same, are they calculated using all available cells of multiple embryos at all time points? How are the NBT+ cells (Figure 5g) projected onto the existing embedding?

Response 3.12:

The t-SNE embedding is calculated for all samples at 48 h first. Other time points and partial data is re-embedded in the following way. The features of the new point cloud are normalised using the normaliser of the original embedding (at 48h). Then we identify the nearest neighbours of every new point in the existing feature space using kd tree search and assign its coordinate in the embedding to the new point. We then run the embedding again for a few epochs to anneal the new points into the existing embedding. This is done to conserve the original embedding and to save computational time. The final embedding is calculated on all used samples first and kept the same in figures 1,2,3,5.

These days UMAP is typically used and preferred, especially in scRNAseq studies, it could be good to have a supplementary figure comparing the t-sne with UMAP representation

Response 3.13:

We agree that both UMAP and t-SNE are widely used for visualising high-dimensional data, and in many cases they produce comparable results. In our case, we chose t-SNE primarily because the openTSNE implementation offered more extensive documentation and greater flexibility at the time of analysis — particularly in supporting re-embedding of new data onto existing projections, which was important for our workflow (e.g. Fig. 5).

We view the choice between UMAP and t-SNE as largely a matter of preference, given that neither method is used for quantitative analysis in this study. Our conclusions are based on the structure of the full feature space, and not on the 2D embedding. We are obviously willing to provide an additional figure comparing the two embeddings but we're not sure that this would substantially improve the manuscript.

Github. We appreciate the efforts of the authors to openly share the code. However, the code is segmented over multiple repositories, has limited readme's and no example data. We recommend:

The readme's could be improved, please add: dependencies (numpy, pandas, matplotlib, tqdm, vedo, scipy, scikit-learn, seaborn, openTSNE, tiffle, h5py, etc), instructions for installation, etc. The entire nuQLOUD pipeline consists of many steps. Would it be possible to provide a small data (crop), and have a notebook that shows the entire pipeline: segmentation, Voronoi, feature extraction, archetype classification?

We are unable to test the nuQLOUD example notebook, since the "raster_geometry" module (used in IntensityMapping.py) is unknown.

Response 3.14:

We thank the reviewer for these constructive suggestions and have substantially updated and extended the GitHub repositories in response.

Specifically:

- We improved the documentation by expanding the readme to include a complete list of dependencies and structured installation instructions, as well as guidance on usage, and contribution.
- We now provide an example dataset consisting of a single segmented zebrafish embryo, along with an accompanying Jupyter notebook that demonstrates the full nuQLOUD pipeline, including feature extraction, normalisation, clustering, and identification of organisational motifs and archetypes.
- The raster_geometry module used in IntensityMapping.py is publicly available via PyPI and can be installed using 'pip install raster-geometry'. We have clarified this in the documentation.

Due to file size limitations on GitHub, the full dataset used in the manuscript cannot be hosted directly, but we are happy to provide access to it upon request. We hope these improvements make the pipeline more accessible and reproducible for the community.

Minor comments:

The figures and videos definitely undersell the work. Much more effort should be put into making the figures a bit more clear and even more work is required for the videos to really showcase the very interesting results of the paper.

We appreciate the reviewer's interest in the clarity and communication of our results. We have invested considerable effort into the design of the figures and videos, and feedback from other reviewers and colleagues has been positive. That said we are happy to further support accessibility of the data. In response to this comment, we now provide 3D visualisations of whole-embryo point clouds as interactive supplementary media, viewable in any standard web browser. These allow readers to explore the spatial organisation and classification results directly and may complement the static figures in illustrating the full scope of the dataset.

We remain open to incorporating additional improvements in future versions of the manuscript and welcome suggestions.

Please add Wan et al. (Cell 2019), Lange et al (Cell, 2024), Amat et al. (Nature Methods, 2014), and Moyle et al. (Nature, 2021) as citations to the statement "in toto imaging ... large-scale tracking ... during zebrafish gastrulation" in order to fairly cover the contributions in the field.

We thank the reviewer for this helpful suggestion and agree that these studies represent important contributions to the field of in toto imaging and large-scale tracking in zebrafish. We have now added the suggested citations to the relevant sentence in the revised manuscript.

Out of interest, does the Voronoi tessellation in any way match the actual shape of the cells? Because one of the features is the number of neighbors, I am wondering whether the number of Voronoi neighbors matches the number of actual touching neighbors of the cell.

In general, the Voronoi tessellation does not match the actual shape of cells, and the number of Voronoi neighbours does not directly correspond to the number of physically contacting neighbours. Rather, it provides a geometrically consistent estimate of spatial neighbourhood relationships based on nuclear positions alone, which is sufficient for our analysis of local organisation. While this approximation abstracts away detailed cell morphology, it captures key spatial constraints and has been widely used for modelling and feature extraction in point cloud-based representations.

Can the authors comment on whether the expensive Voronoi tessellation is necessary? Can you imagine a set of features purely calculated from the point cloud that allows extraction of the same classes?

We thank the reviewer for raising this point. We consider the Voronoi tessellation a necessary component of our analysis, as it provides a principled and scalable way to define local neighbourhoods and extract geometric features such as neighbour count, and shape anisotropy — all of which are critical for distinguishing organisational motifs. We use the voro++ library (Rycroft et al. 2009) for this purpose, which performs the computations efficiently even at large scale.

Importantly, the Voronoi tessellation is calculated directly from the 3D point cloud and does not rely on any additional data structures or external labels. While it is theoretically possible to estimate certain features from the raw point cloud using alternative approaches (e.g. fixed-radius neighbourhoods or kernel density estimates), these would still require computational effort to impose structure on the data. In our view, the Voronoi diagram provides a mathematically grounded and consistent way to achieve this, and we see it as comparable in complexity to other standard methods for deriving neighbourhood relationships in unstructured data.

The authors use t-SNE and UMAP embeddings in the manuscript. Can the authors comment on why they sometimes choose UMAP over t-SNE, and why t-SNE is chosen for the vast amount of embeddings in the manuscript? Also, please comment on the stochastic nature of t-SNE (vs. PCA for example), are the results affected by the t-SNE giving different outputs when repeatedly applied to the same data?

t-SNE and UMAP are two different methods that can embed features of high dimensional data into lower dimensions (here 2D for plotting). We use t-SNE embedding for work with nuQLOUD data for the added flexibility that the package open t-SNE provides (see above). Likewise, we chose UMAP embedding for representing scRNAseq data, since the tools used in the study (scanpy) readily allowed us to do so. Embeddings in this study are purely used for illustrational purposes and no analysis is done using the coordinates of cells in lower dimensional space. All analysis is either done on the full feature set (nuQLOUD) or on the first 50 principal components of the data (scRNAseq, following common practices). t-SNEs are initialised via PCA to limit the effects of stochasticity on the output.

Can the authors comment on the ignored dynamics of the cells? Are the cell movement dynamics considered slow enough that the amorphous/crystalline structure are constant? Can the authors give an estimate for the speed of cell dynamics, relative to the camera integration time?

We thank the reviewer for this thoughtful question. Our analysis is based entirely on fixed samples, and thus we do not directly measure cell movement dynamics. However, we believe the organisational archetypes we observe represent stable states rather than transient configurations for two main reasons:

1. Consistency across time-matched samples: If tissues were fluctuating rapidly between archetypes, we would expect to observe sample-to-sample variability at a given time point. Instead, we find that tissues consistently fall into the same archetype across multiple embryos at matched developmental stages (see Supplementary Fig. 4).

2. Developmental progression: We observe a gradual and consistent emergence of archetypes across time, rather than erratic or unstable patterns. This suggests that the organisational states are developmentally regulated and not dominated by fast, reversible fluctuations.

Because we work with fixed tissues, there is no integration time or motion blur in the imaging. That said, we acknowledge that some tissues may undergo reorganisation on intermediate timescales — an interesting possibility we hope to explore in future live imaging studies.

The submission package contains 3 SI videos, however the videos are not mentioned in the manuscript. Also, volume rendering is probably more interesting than the current slice-by-slice view. This can be done rather easily using Napari or any other similar tool that supports 3D rendering.

We thank the reviewer for the suggestion. The videos are referenced in the main text: "... which confirmed that muscle cells inside each forming somite downregulate N-cad expression in an anteroposterior 'wave-like' manner (Fig. 4d, Suppl. Movies 1–3)."

Regarding the format of the movies: we chose a slice-by-slice view to allow readers to inspect the internal 3D distribution of N-cadherin expression through the depth of the tissue. While volume rendering can produce visually striking images, it often obscures internal features — especially subtle expression gradients — which are central to the interpretation of these data. We therefore opted for a format that prioritises transparency and interpretability over visual effect.

Typos/captions/figures:

We are grateful to the reviewer for carefully highlighting these issues.

"Muscles depends on their being bundled" > remove "their" (introduction)

This is intended, as native-level English speakers may likely expect a reference to what is being bundled. We could write 'these cells being bundled' but that's clunky, 'their' is a standard way to achieve this, this can be ignored by other readers. We'll obviously remove this word if necessary.

“A stage where the the progenitors” > remove double “the” (results)

Corrected.

“From transgenic zebrafish lines and via hybridisation” > remove “and” (results)

This is intended, these are two different approaches – we quantify gene expression via transgenic lines and hybridisation’. We will clarify this by replacing zebrafish with ‘reporter’.

“Of multiple samples.N: num” > missing space before “N” (caption Figure 1)

Corrected

“Solid lean” > “solid line” (caption SI Figure 4)

Corrected

The red text “segmentation” in panel a of Figure 1: technically the red dots seem to represent the cell centers, not the segmentation

The reviewer is correct, the dots represent the centres of the GMM used to segment the nuclei. However, we believe the word ‘segmentation’ allows for more intuitive interpretation of this illustration.

Figure 1g: the caption does not contain information about what the colored lines represent

We have added a note in the figure caption.

Figure 2b: there is a manuscript-wide inconsistency in whether there is a space before “hpf” (“12hpf” vs. “12 hpf”)

Corrected. We now use 12 hpf consistently.

Figure 2de: the overlay of the two archetypes over the embryos is good as an overview, but not very informative. Can you show zooms of certain areas within the embryo that show clear examples of the amorphous/crystalline nature of the tissue?

This is a nice suggestion. Such zoom-ins can be found in Fig. 4 b. We also added interactive 3D renderings as supplementary media for the reader to explore themselves. Moreover, we added projections of the organisational motifs in Suppl. Fig. 9 which give further intuitive insights into the organisational patterns described in this work.

Figure 2: the caption contains important information that is not present in the Main text. Please make sure the information is present in both (for example that HAC is used as the clustering algorithm)

We added a sentence in the main text clarifying the used clustering algorithm.

Figure 3d: the colors are confusing. Do the pink/green overlays on the t-SNE plots represent the two archetypes or the “cdh2”/“cdh1”?

We added a caption to the plot clarifying this confusion.

Figure 4: the panel indications are not horizontally aligned (for example “c”)

This is due to the layout of the figure and intended.

On multiple occasions in the manuscript, the numerical notion $\text{mean} \pm \text{std}$ has the wrong number of significant figures. For example, “116,600 \pm 9,000” should be “117e3 \pm 9e3” or “(117 \pm 9)e3”, since the std can only have 1 significant figure.

We adjusted the reporting of mean/std accordingly in the text and figures.

Part of the Microscopy section of the Methods has a different vertical line spacing

We corrected the line spacing.

Reviewer #3 (Remarks on code availability):

Th repo requires a lot of work: there is no readme, the dependencies are not specified, and there is overall very little effort to make the work reproducible or even useful for others. This is unfortunate because the rest of the work and manuscript is interesting. Improving the repo and making it truly useful for others would really make a difference and improve the impact and utility of this work for others.

We thank the reviewer for their critical assessment and fully agree that improving accessibility and reproducibility is crucial for maximising the impact and utility of our work. We had already recognised these issues based on earlier reviewer comments and have made substantial improvements to the GitHub repositories during the revision process.

Specifically:

- We significantly expanded the documentation, including a detailed README with a full list of dependencies, installation instructions, usage examples, and contribution guidelines.
- We now provide a complete example dataset (one segmented zebrafish embryo at 48 hpf) alongside a Jupyter notebook that runs through the full nuQLOUD analysis pipeline,

covering segmentation input, Voronoi tessellation, feature extraction, archetype classification, and visualisation.

- While the full original dataset is too large to host directly on GitHub, we offer access to it upon request.

We hope that these improvements have addressed the reviewer's concerns and substantially enhanced the usability and reproducibility of the nuQLOUD framework for the community.